# Demystifying Deep Search: A Holistic Evaluation with Hint-free Multi-Hop Questions and Factorised Metrics

**Maojia Song**[*]
Singapore University of Technology and Design (SUTD)
maojia_song@mymail.sutd.edu.sg

**Renhang Liu**[*]
Nanyang Technological University (NTU)
renhang.liu@ntu.edu.sg

**Xinyu Wang , Yong Jiang, Pengjun Xie, Fei Huang**
Tongyi Lab, Alibaba Group
{xinyu.wxy, jiangyong.jy, xiepengjun.xpj, f.huang}@alibaba-inc.com

**Soujanya Poria**[†]
Nanyang Technological University (NTU)
soujanya.poria@ntu.edu.sg

**Jingren Zhou**
Tongyi Lab, Alibaba Group
jingren.zhou@alibaba-inc.com

## Abstract

RAG (Retrieval-Augmented Generation) systems and web agents are increasingly evaluated on multi-hop deep search tasks, yet current practice suffers from two major limitations. First, most benchmarks leak the reasoning path in the question text, allowing models to follow surface cues rather than discover reasoning chains autonomously. Second, evaluation is typically reduced to a single pass rate, which collapses diverse behaviours into one score and obscures whether failures stem from inadequate search, poor knowledge use, or inappropriate refusal. To address these issues, we present **WebDetective**, a benchmark of hint-free multi-hop questions paired with a controlled Wikipedia sandbox that ensures full traceability of model actions, and a holistic evaluation framework that separates search sufficiency, knowledge utilisation, and refusal behaviour. Our evaluation of 25 state-of-the-art models reveals systematic weaknesses across all architectures: models struggle with knowledge utilisation despite having sufficient evidence and demonstrate near-absent appropriate refusal when evidence is lacking. These patterns expose a fundamental gap—today's systems excel at executing given reasoning paths but fail when required to discover them. We develop an agentic workflow **EvidenceLoop** that explicitly targets the challenges our benchmark identifies, incorporating verification loops and systematic evidence tracking that improve both search and synthesis capabilities. This baseline demonstrates that **WebDetective**'s diagnostic framework can guide concrete architectural improvements, establishing our benchmark as a critical tool for developing genuinely autonomous reasoning systems rather than pattern-following agents.

## 1 Introduction

Web agents—autonomous systems that navigate and extract information online—extend language models beyond parametric knowledge by combining search with internal reasoning, aggregating evidence, and synthesising answers Nakano et al. (2021); Ferrag et al. (2025). A core evaluation setting for these systems is *deep search*: multi-step discovery under noise, hypothesis generation, and the persistent "I can't find it" barrier faced even by expert searchers. Recent work improves agent designs Li et al. (2025b;c); Sun et al. (2025); Jiang et al. (2025); Li et al. (2025a) and proposes new benchmarks Du et al. (2025); Wong et al. (2025).

---

[*]Equal Contributions.
[†]Corresponding author.

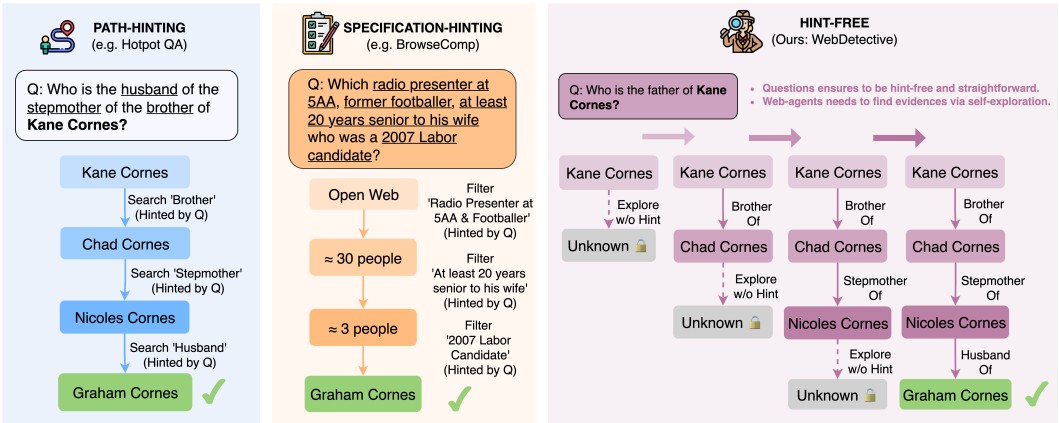

Figure 1: Comparison of different question formulations in multi-hop deep search. **Left:** Path-Hinting (PH) benchmarks such as HotpotQA embed the reasoning path directly in the question text, effectively reducing reasoning to execution. **Middle:** Specification-Hinting (SH) benchmarks such as BrowseComp obscure the target entity behind multiple attributes, testing filtering rather than autonomous exploration. **Right:** Our Hint-Free (HF) formulation in **WebDetective** removes both path and specification hints, requiring agents to autonomously discover reasoning chains within a controlled Wikipedia sandbox.

However, we identify a largely overlooked issue: many deep-search benchmarks contain *hinting* that steers models toward the answer, reducing the need for genuine autonomous reasoning. Such hints bypass a key capability of web agents: *discovering which connections matter, forming hypotheses about promising reasoning trajectories, and adaptively exploring without guidance*.

As shown in Figure 1, classical multi-hop QA datasets (e.g., HotpotQA Yang et al. (2018a)) exhibit **Path-Hinting (PH)**, where the question narrates the reasoning chain (e.g., "Who is the husband of the stepmother of the brother of Kane Cornes?"), turning the task into executing a prescribed path. Newer benchmarks (e.g., BrowseComp Wei et al. (2025) and WebShaper Tao et al. (2025)) introduce **Specification-Hinting (SH)**, describing the target via distinctive attributes. While less explicit, SH often becomes constraint filtering rather than exploratory search. In both cases, agents receive scaffolding uncommon in real use, limiting evaluation of true autonomy.

A second limitation is that evaluations typically report only aggregate pass rates, collapsing distinct failure modes. An agent that searches well but fails to synthesise evidence differs fundamentally from one that stops early, over-relies on parametric knowledge, or refuses when it should continue. When evidence is insufficient, models should recognise and signal uncertainty rather than guess. Without separating these behaviours, diagnosis and progress are difficult.

We introduce **WebDetective**, a benchmark that co-designs *hint-free* questions and a controlled environment for fine-grained diagnostics. We construct **Hint-Free (HF) Multi-Hop** questions that avoid both path narration and attribute fingerprints while retaining a unique answer (e.g., "Who is the father of Kane Cornes?"). Questions are paired with a controlled Wikipedia sandbox that includes a reference reasoning path. To prevent shortcutting, each node in the chain is masked so the next entity is reachable only through its predecessor. This supports comparison between arbitrary valid strategies and a canonical shortest search-only path, while disentangling search sufficiency, synthesis, and final success. We also propose an agentic baseline workflow, **EvidenceLoop**, with explicit context retention, memory management, and verification steps.

Using our diagnostic framework, we evaluate 25 representative models and observe a consistent pattern: without hints, even strong systems succeed on only a limited subset, and their failures differ from what conventional benchmarks suggest. Many retrieve relevant evidence but fail to assemble a coherent reasoning chain; others abandon search prematurely or produce unsupported answers despite clear knowledge gaps, indicating weak calibrated refusal. These results highlight the need to evaluate *path discovery* rather than path execution.

Our contributions are:

- **WebDetective**, the first benchmark to systematically remove both path and specification hinting while enabling controlled, fine-grained diagnostics.

- An evaluation framework that separates search sufficiency, information synthesis, and refusal behaviour to pinpoint failure modes in deep-search agents.

- **EvidenceLoop**, an agentic workflow with context retention, memory management, and verification, demonstrating how the benchmark can guide concrete system improvements.

## 2 THE **WEBDETECTIVE** BENCHMARK

### 2.1 HINT-FREE MULTI-HOP QUESTION ANSWERING

Given a question $q$ and a knowledge corpus $\mathcal{C}$, an agent must find the answer $a^*$ by discovering and composing a sequence of evidence pieces $\mathcal{E} = \{e_1, \ldots, e_n\}$ from $\mathcal{C}$. Each $e_i$ is an atomic fact on the page of entity $v_i$ that links (semantically or relationally) to $v_{i+1}$, forming a reasoning chain $v_0 \to v_1 \to \cdots \to v_n$, where $v_0$ is the starting entity (typically mentioned in $q$) and $v_n$ yields $a^*$. The reasoning function $\mathcal{R}_{\text{func}} : \mathcal{E} \to a^*$ specifies how these facts are composed—via logical inference, relation transitivity, or domain rules. We call any information embedded in $q$ that reveals $\mathcal{R}_{\text{func}}$ or fingerprints $v_n$ a *hint* $h$.

We define two prevalent hint types in existing multi-hop QA benchmarks Yang et al. (2018b); Chen et al. (2019); Wei et al. (2025):

**Path-Hinting (PH):** The question directly encodes the reasoning chain, with $h_{\text{PH}} = \text{Encode}(\mathcal{R}_{\text{func}})$ revealing its structure. For example, "Who is the husband of the stepmother of the brother of Kane Cornes?" decomposes into: find brother $\to$ stepmother $\to$ husband. The agent no longer discovers the path, but executes the specified $h_{\text{PH}}$.

**Specification-Hinting (SH):** The question hides the target behind constraints, with $h_{\text{SH}} = \{s_1, s_2, \ldots, s_k\}$ narrowing the search space to a unique entity. For instance, "Which radio presenter at 5AA, former footballer, at least 20 years senior to his wife who was a 2007 Labor candidate?" yields $h_{\text{SH}} = \{$radio presenter at 5AA, former footballer, 20+ years senior to wife, wife was 2007 Labor candidate$\}$, collectively fingerprinting Graham Cornes; the task reduces to constraint satisfaction (match all $s_i$) rather than discovering which connections matter for reasoning.

In contrast, we propose **Hint-Free (HF) Multi-Hop** QA, where $h = \emptyset$. Questions contain only the essential query, without path narration or attribute fingerprints—for example, "Who is the father of Kane Cornes?" Answering requires discovering evidence pieces $e_1$ (Kane Cornes has brother Chad Cornes), $e_2$ (Chad's stepmother is Nicole Cornes), $e_3$ (Nicole's husband is Graham Cornes), and composing them via familial reasoning to derive $a^* = $ Graham Cornes. Crucially, the agent must independently uncover both the evidence chain $\mathcal{E}$ and the reasoning function $\mathcal{R}_{\text{func}}$, capturing the fundamental ability to transform a simple information need into an autonomously constructed reasoning structure.

### 2.2 THE CO-DESIGN PRINCIPLE

While hint-free questions eliminate linguistic scaffolding, we observe that question design alone is insufficient to ensure genuine multi-hop reasoning. In open corpora or live web environments, even well-designed hint-free questions permit shortcuts that bypass the intended reasoning chain. Consider our example "Who is the father of Kane?"—in Wikipedia or web search, direct co-occurrences of "Kane" and "Graham Cornes" may exist in unrelated contexts, or intermediate entities like "Chad Cornes" could be found through direct search, allowing agents to bypass the intended reasoning path. Moreover, because both answers and intermediates are usually accessible through multiple paths, it becomes impossible to tell whether an agent truly discovered the reasoning chain or simply relied on shortcuts or prior knowledge.

$$v_i \text{ is discoverable} \iff \text{agent visits page}(v_{i-1})$$

This masking eliminates *corpus-based shortcuts*—agents cannot find the answer through spurious co-occurrences or unrelated contexts—while still accommodating diverse valid reasoning strategies. The ground-truth chain serves as a **reference path** (the shortest sufficient route using the corpus alone), but our evaluation framework does **not** enforce strict adherence to this path. To comprehensively assess whether an agent's search is sufficient and efficient, we employ a multi-faceted evaluation:

**(1) Shortest Reference Chain Coverage.** We first check whether all hops in the reference chain were visited. For any missing evidence, we probe the model's parametric knowledge with targeted queries (e.g., "Kane Cornes has brother ___?"). If visited evidence plus verified parametric knowledge covers the complete chain, the instance achieves knowledge sufficiency.

**(2) Alternative Path Recognition.** Agents may discover valid reasoning routes different from the reference chain. We collect *all* evidence the agent actually visited during search and feed this clean context to an LLM judge. If the judge can correctly answer the question from this visited evidence alone, the instance also achieves knowledge sufficiency—regardless of whether the reference chain was followed. Consider one real example from our benchmark: "On which compilation album does the song 'Loneliness of a Middle Distance Runner' appear?" The reference path follows: Novel (The Loneliness of the Long-Distance Runner) → Single (Jonathan David) → Compilation (Push Barman to Open Old Wounds), tracing through the single that featured the song as a B-side. However, an agent might instead traverse Novel → Belle and Sebastian (the band mentioned as adapting the title) → the band's discography page listing "Push Barman to Open Old Wounds" as a B-sides compilation—reaching the same answer through the artist's discography rather than the specific single.

**(3) Efficient Parametric Knowledge Use.** Our **Search Score** further credits agents that efficiently combine search with parametric knowledge. When an agent correctly answers the question using fewer (or equal) hops than the reference path while actively performing search, this demonstrates intelligent leveraging of internal knowledge to shortcut the reasoning process, and receives additional credit.

Returning to our running example, "Who is the father of Kane Cornes?", the reference path is Kane → Chad (brother) → Nicole (stepmother) → Graham (husband). An agent achieves knowledge sufficiency by: (a) visiting all four pages, (b) visiting partial pages while parametric probes confirm missing knowledge, or (c) discovering alternative evidence that supports the correct answer. An agent that correctly answers using only two hops (e.g., Kane → Chad, then leveraging parametric knowledge about Chad's family) receives Search Score credit for efficient reasoning. This design eliminates unearned shortcuts while rewarding genuine multi-hop reasoning through any valid strategy.

Lastly, while we instantiate this benchmark using Wikipedia as a controlled testbed, the co-design methodology is not inherently tied to this corpus. The framework generalises to any domain satisfying three conditions: (1) a text corpus with factual content, (2) an entity-level graph or link structure over that corpus, and (3) the ability to mask entity mentions along chosen paths. Given an appropriate dataset meeting these requirements, the same evaluation pipeline—path construction, selective masking, and factorised metrics—can be directly applied to news archives, scientific repositories, enterprise knowledge bases, or curated web snapshots, extending the diagnostic value beyond the current Wikipedia instantiation.

## 2.3 BEYOND PASS RATE: A DIAGNOSTIC EVALUATION FRAMEWORK

Traditional multi-hop QA evaluation reduces performance to a single pass rate, obscuring distinct failure modes: an agent that searches exhaustively but fails to synthesise evidence is fundamentally different from one that refuses prematurely or hallucinates from parametric knowledge. Our sandbox, with guaranteed unique reasoning paths, enables precise diagnosis by separating *knowledge sufficiency* (whether the agent has obtained the necessary evidence through search or memory) from *generation quality* (whether it can synthesise a correct answer or appropriately refuse). This decomposition reveals that similar pass rates can mask very different underlying capabilities. See Appendix A for details.

**Knowledge Discovery Metrics.** We assess whether agents acquire necessary information through two complementary metrics. **Knowledge Sufficiency** determines if an agent possesses all required evidence $\mathcal{E} = \{e_1, ..., e_n\}$ for answering either from search or parametric knowledge. We track which evidence the agent discovered through search by monitoring visited pages in our sandbox. For any missing evidence $e_i \notin \mathcal{E}_{\text{found}}$, we probe the model's parametric knowledge with targeted queries (e.g., "Kane Cornes has brother ____?"). The **Search Score** extends this by crediting models that efficiently combine partial search with parametric knowledge—recognizing that if an entity discovered through search has a meaningful relationship to the answer stored in parametric memory, this represents legitimate reasoning that demonstrates efficient knowledge utilization.

**Generation Quality Metrics.** Conditioned on knowledge sufficiency, we classify cases as knowledge-sufficient ($\mathcal{S}$) or insufficient ($\mathcal{I}$), and attempted ($\mathcal{A}$) or refused ($\mathcal{N}$). This yields two key metrics. **Good Refusal (GR)** evaluates abstention when evidence is lacking, with $\text{F1}_{\text{GR}}$ balancing recall (avoiding hallucination) and precision (refusing only when justified). **Knowledge Utilisation (KU)** measures synthesis when evidence is present, with $\text{F1}_{\text{KU}}$ balancing recall (using available evidence) and precision (answers grounded rather than speculative). These complementary F1 scores capture refusal discipline and evidence integration.

We define a unified **Generation Score** as

$$\text{GenScore} = \frac{\text{F1}_{\text{GR}} + \text{F1}_{\text{KU}}}{2} \cdot \text{KnowledgeScore},$$

where **Knowledge Score** is the fraction of instances with knowledge sufficiency (all evidence gathered). The sufficiency weighting prevents gaming by agents that simply refuse all questions, ensuring models are rewarded only when they both acquire the necessary evidence and handle it appropriately—through correct synthesis or justified refusal.

**Knowledge Degradation Analysis.** Even when models achieve knowledge sufficiency, they often fail to generate correct answers, revealing a gap between evidence possession and synthesis. To diagnose this gap, we design two tests. The **Knowledge Forget** test captures cases where models fail to apply parametric knowledge in full-context reasoning, despite succeeding on isolated probes. The **Lead-astray** test captures failures caused by noisy search contexts—irrelevant pages, failed attempts, or exploration clutter—that prevent models from synthesising answers they could otherwise produce from clean evidence. Together, these metrics go beyond simple pass rates by pinpointing whether errors stem from inadequate search, over-confident hallucination, weak synthesis, or degraded reasoning under noise, thereby clarifying the capabilities required for robust multi-hop reasoning.

## 2.4 DATASET CONSTRUCTION

We build **WebDetective** through a pipeline that transforms single-hop Wikipedia QA pairs into verified multi-hop reasoning chains, ensuring every hop is necessary. See Appendix B and Appendix C for details.

**Source Data and Chain Discovery.** Starting from Wikipedia QA pairs where each question $q$ references a starting entity $v_0$ and has an answer $v_n$, we remove the direct link $v_0 \to v_n$ and use BFS on the hyperlink graph to find the shortest alternative path $v_0 \to v_1 \to \cdots \to v_n$. For each edge $(v_i, v_{i+1})$, we extract the sentence $e_i$ in the page of $v_i$ that links $v_i$ and $v_{i+1}$, forming the evidence chain $\mathcal{E} = \{e_1, \ldots, e_n\}$.

**Verification of Necessity.** Since many graph paths are irrelevant, we apply three automated checks with a strong LM (Qwen-3-235B-A22B): (1) *Parametric Inaccessibility*: $\text{LM}(q) \neq v_n$, ensuring the answer is not retrievable from parametric memory alone. (2) *Evidence Sufficiency*: $\text{LM}(q, \mathcal{E}) = v_n$, confirming the full chain supports the answer. (3) *Evidence Necessity*: $\text{LM}(q, \mathcal{E} \setminus \{e_i\}) \neq v_n$ for all $e_i$, verifying no hop is redundant.

**Human Validation.** Surviving questions are manually checked to ensure all evidence is required, reasoning is logically sound, and no hints are embedded in the wording. The final benchmark contains 200 validated questions with varied hop counts and types.

**Domain Coverage.** The final dataset spans diverse domains to ensure broad evaluation coverage: Film, Television & Theatre (20.0%), History, Politics & Military (18.0%), Geography & Landmarks

(16.0%), Music (13.5%), Sports (11.5%), Biography & Education (8.5%), Literature & Pop Culture (6.5%), and Science & Technology (6.0%). This distribution ensures WebDetective is not skewed toward any single relation type but covers reasoning patterns essential for evaluating multi-hop capabilities.

## 3 A BASELINE ATTEMPT: THE EVIDENCELOOP FRAMEWORK

To address the unique challenges posed by hint-free multi-hop reasoning, we develop **EvidenceLoop**, an agentic workflow baseline that explicitly incorporates context retention, memory management, and verification steps to maintain reasoning coherence across extended search trajectories. Unlike standard ReAct implementations that can lose track of evidence across many search iterations, our workflow introduces structured mechanisms for tracking discovered entities, maintaining evidence chains, and verifying reasoning paths before answer generation.

**Iterative Refinement with Fallback.** Our framework balances exploration breadth with computational efficiency through $R_{\max}$ iterations. In each round $r$, $N$ solver agents explore different reasoning paths in parallel, each operating with up to $B$ actions and guided by an aggregated context $C^r$ ($C^0 = \emptyset$). Their outputs are refined through two stages: (1) an *extraction agent* distills key findings, entity references, and promising paths; and (2) an *aggregation agent* synthesises them into $C^{r+1}$, retaining valuable evidence while discarding noise. This process allows early iterations to explore broadly—sports connections, geographic links, family ties—while progressively focusing on the most promising directions. If no conclusive answer is found after $R_{\max}$ rounds, a final fallback stage consolidates all evidence into $C^{\text{final}}$ and invokes a synthesis-only solver, distinguishing failures due to insufficient exploration from those due to poor evidence composition.

**Evidence Memory System.** Iterative refinement is supported by a persistent memory $\mathcal{M}$ that records all evidence retrieved during search. Each piece of evidence is assigned a unique Evidence ID (EID) and stored with its full content. Agents receive both summaries and EID references (e.g., "Kane has brother Chad [EID-042]"), enabling them to reason over concise contexts while retaining the ability to fetch full documents on demand through a retrieve action. This dual representation prevents agents from being overwhelmed by long documents or lossy compressions: they work with focused summaries but never lose access to the complete evidence trail. EIDs also enable systematic traceability, as later modules can verify claims against their original sources.

**Verification.** To ensure evidence-grounded reasoning, any proposed answer must be decomposed into atomic claims $\{c_1, \ldots, c_m\}$, each linked to an EID. A verification agent $V$ checks that (1) each claim is entailed by the source content, (2) the claims collectively support the proposed answer, and (3) the answer directly addresses the question. Verification occurs during execution: rejected proposals return specific feedback, allowing solvers to repair reasoning within the remaining budget $B$, while accepted answers immediately terminate exploration. This mechanism prevents hallucinations from propagating, enforces tight evidence grounding, and improves efficiency by halting search as soon as verification succeeds.

**Positioning Relative to Existing Frameworks.** **EvidenceLoop** is a diagnostically-derived workflow designed to address the specific failure modes revealed by **WebDetective**. Table 1 summarises the key architectural differences from standard approaches. We provide a more detailed description of the **EvidenceLoop** architecture, including its controller configuration, memory modules, and verification procedures in Appendix E.

Table 1: Architectural comparison of **EvidenceLoop** with existing agentic frameworks (Standard ReAct and ReAct with Reflection).

| Aspect | Standard ReAct | ReAct + Self-Reflection | EvidenceLoop (Ours) |
|---|---|---|---|
| Controller Loop | Single thought-tool-observation | Same + occasional reflection | Two-phase: exploration then verification |
| Memory | Flat dialogue history | Same + reflection messages | Structured buffer: (entity, snippet, source) tuples |
| Verification | Often none | Critiques not tied to evidence | Explicit loop using only evidence buffer |
| Breadth & Iterations | Single trajectory | Single trajectory | Parallel trajectories + explore-aggregate cycles |
| Refusal Behaviour | Implicit, rarely triggered | Loosely specified | Explicitly tied to verification: incomplete $\rightarrow$ refuse |

Table 2: Comparison of 25 state-of-the-art models with ReAct-style tool use capabilities. Metrics cover Knowledge Discovery (Knowledge Sufficiency, Search Score), Generation Quality (Generation Score, Good Refusal F1, Knowledge Utilisation F1), Knowledge Degradation (Forget, Lead-astray), and Pass@1. **Bold** values denote best results: higher is better for Knowledge Discovery, Generation Quality, and Pass@1, while lower indicates greater robustness for Knowledge Degradation. EvidenceLoop uses DeepSeek-R1 as its base model; shaded rows highlight this comparison pair.

| Provider | Model | Knowledge Discovery | | Generation Quality | | | Knowledge Degradation | | Pass@1 |
| | | Knowledge Suff. (%) | Search Score (%) | Generation Score (%) | Good Refusal F1 (%) | Knowledge Util. F1 (%) | Forget (%) | Lead-astray (%) | (%) |
|---|---|---|---|---|---|---|---|---|---|
| OpenAI | GPT-OSS-120B OpenAI (2025b) | 16.00 | 23.50 | 2.75 | 23.59 | 10.73 | 100.00 | **0.00** | 24.00 |
| | o3-Mini OpenAI (2025c) | 48.50 | 57.00 | 9.10 | 21.05 | 16.48 | 46.39 | 42.27 | 21.50 |
| | o4-Mini OpenAI (2025d) | 68.00 | 72.00 | 12.69 | 19.75 | 17.56 | 27.94 | 59.56 | 21.00 |
| | o3 OpenAI (2025c) | 70.00 | 76.00 | 18.29 | 3.29 | 48.97 | 24.29 | 24.29 | 53.50 |
| | o3-Pro OpenAI (2025c) | 71.00 | 78.00 | 20.86 | 9.37 | 49.40 | 21.83 | 25.35 | **56.00** |
| | GPT-5-Chat OpenAI (2025a) | 58.00 | 59.50 | 15.74 | 26.23 | 28.05 | 47.41 | 31.90 | 29.50 |
| | GPT-5 OpenAI (2025a) | **79.00** | **80.00** | 23.21 | 8.89 | 49.58 | **17.72** | 32.91 | 50.50 |
| Anthropic | Claude-Sonnet-4-Think Anthropic (2025) | 66.50 | 73.50 | 26.19 | 34.59 | 44.19 | 45.11 | 21.80 | 38.50 |
| | Claude-Opus-4-Think Anthropic (2025) | 68.00 | 73.50 | 21.00 | 30.53 | 31.23 | 43.38 | 32.35 | 29.00 |
| | Claude-Opus-4.1 Anthropic (2025) | 74.00 | 76.50 | 28.53 | 28.57 | 48.54 | 27.03 | 31.08 | 44.50 |
| Google | Gemini-2.5-Flash-Think Google DeepMind (2025) | 59.00 | 64.50 | 16.79 | 40.56 | 16.35 | 57.63 | 35.59 | 17.50 |
| | Gemini-2.5-Pro Google DeepMind (2025) | 65.50 | 73.00 | 11.64 | 10.87 | 24.68 | 44.27 | 35.11 | 28.50 |
| xAI | Grok-4 xAI (2025) | 74.00 | 77.50 | **34.71** | 37.63 | **56.19** | 23.65 | 27.70 | 50.50 |
| Alibaba | Qwen3-30B-Think Yang et al. (2025) | 56.50 | 59.00 | 7.25 | 12.51 | 13.16 | 79.65 | 16.81 | 12.50 |
| | Qwen3-235B-Think Yang et al. (2025) | 72.50 | 72.00 | 11.15 | 6.56 | 24.19 | 63.45 | 19.31 | 21.50 |
| | Tongyi-DeepResearch Tongyi DeepResearch Team (2025) | 53.50 | 57.50 | 4.20 | 0.00 | 15.69 | 43.93 | 41.12 | 18.50 |
| ByteDance | Doubao-1.6-Flash ByteDance Seed Team (2025) | 54.50 | 57.50 | 20.00 | **53.95** | 19.46 | 68.81 | 21.10 | 13.50 |
| | Doubao-1.6-Think ByteDance Seed Team (2025) | 64.00 | 68.50 | 19.24 | 42.03 | 18.11 | 49.22 | 30.84 | 16.00 |
| Zhipu AI | GLM-4.5-Air-Inner Zhipu AI Team (2025) | 55.50 | 60.50 | 12.31 | 26.39 | 17.97 | 44.14 | 40.54 | 19.00 |
| | GLM-4.5-Inner Zhipu AI Team (2025) | 63.50 | 67.50 | 22.19 | 34.79 | 35.09 | 25.98 | 40.16 | 33.50 |
| Moonshot AI | Kimi-K2-0711 Moonshot AI (2025) | 54.50 | 59.00 | 9.72 | 16.36 | 19.31 | 43.12 | 36.70 | 23.50 |
| | Kimi-K2-0905 Moonshot AI (2025) | 53.00 | 55.00 | 13.17 | 28.79 | 20.89 | 49.06 | 33.96 | 24.00 |
| DeepSeek | DeepSeek-R1 DeepSeek-AI et al. (2025) | 61.50 | 65.50 | 10.57 | 18.81 | 15.55 | 37.40 | 51.22 | 20.00 |
| | DeepSeek-V3.1 DeepSeek-AI et al. (2024) | 67.50 | 56.50 | 13.62 | 27.97 | 16.34 | 44.72 | 44.72 | 17.00 |
| | DeepSeek-V3.1-Terminus DeepSeek-AI et al. (2024) | 55.50 | 58.50 | 16.31 | 36.49 | 22.23 | 28.83 | 50.45 | 24.50 |
| Our Team | EvidenceLoop | 61.50 | 62.50 | 12.61 | 17.98 | 23.79 | 41.46 | 41.46 | 25.00 |

## 4 EXPERIMENTS

We evaluate 25 state-of-the-art models with ReAct-style tool use, including those from OpenAI, Anthropic, Google, xAI, Alibaba, ByteDance, Zhipu AI, Moonshot AI, and High-Flyer. All models interleave reasoning, search, and observations in our controlled Wikipedia sandbox on **WebDetective** with 200 hint-free multi-hop questions (2–4 hops), under limits of 40 tool calls and a 32K-token context. Unless noted, decoding uses `temperature=0.6`, `top_p=0.95`. We report six metrics: (1) *Knowledge Score*, knowledge sufficiency; (2) *Search Score*, retrieval effectiveness; (3) *Generation Score*, weighted F1 of Good Refusal and Knowledge Utilization; (4) *Good Refusal F1*; (5) *Knowledge Utilization F1*; and (6) *Pass@1*. We additionally analyse *Forget* and *Lead-astray* to probe knowledge degradation (Section 4.2.2). For our **EvidenceLoop**, we use DeepSeek-R1 as the base model and set `breadth=3, iteration=3`. Comprehensive results appear in Table 2.

### 4.1 MAIN RESULTS

**Frontier models are far from saturating the task.** Even the strongest systems reach only ∼50% Pass@1 on our benchmark: o3-Pro tops out at 56.0%, while GPT-5 and Grok-4 both achieve 50.5%; Claude-Opus-4.1 is at 44.5%, and many others fall well below 40%. This illustrates the challenging nature of our benchmark, **WebDetective**.

**Search, generation, and final accuracy are decoupled.** High retrieval does not translate proportionally into better synthesis or Pass@1. For example, GPT-5 attains an 80.0% Search Score but only 23.21% Generation Score and 50.5% Pass@1; o3-Pro similarly has 78.0 Search but 20.86 Generation (56.0% Pass@1). Conversely, Grok-4 achieves the highest Generation Score (34.71) with 77.5 Search and 50.5% Pass@1, while Qwen3-235B-Thinking posts 72.0% Search yet just 11.15% Generation and 21.5% Pass@1. These gaps indicate that *information synthesis, not just retrieval, is a key bottleneck*.

**Refusal ability is underdeveloped.** Good-refusal performance is generally low: the best we observe is 53.95% F1 (Doubao-1.6-Flash). Many frontier models underperform markedly—e.g., GPT-5 (8.89%), o3-Pro (9.37%), and o4-Mini (19.75%)—and even strong generalists like Claude-Opus-

Table 3: **Emergent model profiles from metric interplay analysis.** Models cluster into distinct behavioural profiles based on their knowledge sufficiency, refusal behaviour, and information synthesis, revealing characteristic strengths and failure modes across the spectrum of frontier systems.

| Profile | Metric Pattern | | | Pass@1 | Example Models | Failure Mode |
|---|---|---|---|---|---|---|
| | Knowledge | Refusal | Utilization | | | |
| Powerful but Overconfident | High | Low | High | 50-56% | GPT-5, o3-Pro, o3 | Hallucination from overconfidence |
| Well-Calibrated Elite | High | Med | High | 44-51% | Grok-4, Claude-Opus-4.1 | Minor: unnecessary caution |
| Synthesis Bottleneck | High | Low | Low | 18-22% | Qwen3-235B, Tongyi-DR | Cannot compose multi-hop reasoning |
| Conservative Middle | Med | Med | Med | 29-39% | Claude-Sonnet-4, GLM-4.5 | Under-utilizes capabilities |
| Weak and Confused | Med | Low | Low | 20-22% | o4-Mini, DeepSeek-R1 | Poor synthesis + poor calibration |
| Self-Aware of Weakness | Low | High | Low | 13-18% | Doubao variants, Gemini-Flash | Comprehensive inability (appropriate) |
| Ideal (Unachieved) | High | High | High | – | None | None - optimal behavior |

4.1 remain modest (28.57%). This highlights *weak calibrated abstention* when evidence is insufficient.

**EvidenceLoop demonstrates targeted improvements.** Comparing **EvidenceLoop** to its base model DeepSeek-R1 reveals meaningful gains from our diagnostically-derived design: Pass@1 improves from 20.0% to 25.0% (+25% relative), Generation Score from 10.57% to 12.61% (+19% relative), and most notably, Knowledge Utilization F1 from 15.55% to 23.79% (+53% relative). The substantial improvement in Knowledge Utilization demonstrates that our structured evidence buffer and verification loop directly address the synthesis bottleneck that WebDetective identifies as a critical failure mode. While absolute gains are moderate, these improvements validate that the failure modes exposed by our diagnostic framework can guide concrete architectural improvements. We conduct additional experiments across different base models on **EvidenceLoop** and investigate the effectiveness of each component in Appendix E.

## 4.2 ANALYSIS

### 4.2.1 UNDERSTANDING MODEL FAILURE MODES THROUGH METRIC PATTERNS

To better understand the diverse failure modes in multi-hop reasoning, we analyze the interplay between our three core metrics: Knowledge Sufficiency (ability to gather evidence), Good Refusal F1 (calibration of uncertainty), and Knowledge Utilization F1 (synthesis capability). Rather than examining metrics in isolation, we investigate how their combinations reveal distinct behavioral profiles.

We categorize performance using empirically-derived thresholds: Knowledge Sufficiency (High: $> 70\%$, Medium: 60-70%, Low: $< 60\%$), Good Refusal F1 (High: $> 40\%$, Medium: 25-40%, Low: $< 25\%$), and Knowledge Utilization F1 (High: $> 45\%$, Medium: 25-45%, Low: $< 25\%$). Analyzing all 23 models, we observe that they cluster into six distinct profiles based on these metric combinations, with certain theoretically plausible patterns notably absent from the empirical data.

We present the taxonomy in Table 3. The **Powerful but Overconfident** profile (GPT-5, o3-Pro, o3) achieves the highest pass rates (50-56%) through strong evidence gathering and synthesis, but exhibits dangerous overconfidence with refusal rates below 10% despite 21-30% knowledge insufficiency. These models prefer hallucination over admission of uncertainty. In contrast, the **well-calibrated Elite** (Grok-4, Claude-Opus-4.1) achieves similar knowledge sufficiency and utilisation but maintains moderate refusal rates (29-38%), demonstrating that strong capabilities do not need to preclude epistemic awareness, although this calibration costs approximately 5-6% in pass rate.

The **Synthesis Bottleneck** profile reveals a critical failure mode: models like Qwen3-235B-Thinking achieve high knowledge sufficiency (72.5%) but catastrophically fail at synthesis ($< 25\%$ utilisation). Despite possessing evidence, they cannot compose multi-hop reasoning chains, yet their low refusal rates indicate unawareness of this limitation. The **Conservative Middle** models (Claude-Sonnet-4-Think, GLM-4.5-Inner) exhibit consistent mediocrity across all metrics, suggesting excessive caution—their moderate utilisation (31-44%) despite reasonable knowledge gathering (63-68%) indicates they refuse even when capable of answering.

At the lower performance tiers, we observe a striking divergence in self-awareness. **Self-Aware of Weakness** models (Doubao variants, Gemini-2.5-Flash-Think) appropriately refuse in 40-54% of cases, correctly recognising their limitations in both search and synthesis. Conversely, **Weak and Confused** models (o4-Mini, DeepSeek-R1) exhibit similar capability limitations but fail to recognise them, attempting answers despite 16-18% utilisation rates.

Our analysis reveals three distinct failure modes in the multi-hop reasoning pipeline. *Search failure* affects 21-46% of attempts even in top models, indicating that evidence discovery remains challenging. *Synthesis failure* is more severe—even with sufficient knowledge, utilisation rates peak at 56%, suggesting that composing multi-hop reasoning chains remains a fundamental bottleneck. *Calibration failure* manifests bidirectionally: top-performing models are systematically overconfident (refusing $< 10\%$ despite significant insufficiency), while weaker models may over-refuse or, worse, lack any calibration signal. Notably, no model in our evaluation achieves both high utilisation and high refusal—a perfectly calibrated model would excel at synthesis while maintaining appropriate uncertainty, but current architectures appear to force a tradeoff where strong synthesis capability invariably leads to overconfidence. This suggests a fundamental tension between competence and epistemic humility in existing architectures.

The emergence of these distinct profiles suggests that improving multi-hop reasoning requires targeted interventions. Models in the Synthesis Bottleneck category need architectural improvements to reasoning composition, not better search. Overconfident models need calibration mechanisms that don't sacrifice performance. Most importantly, the absence of any model achieving high performance across all three metrics—even Grok-4 and Claude-Opus-4.1, the best-balanced models, only reaches 50.5% and 44.5% pass rate—demonstrates that robust multi-hop reasoning remains an open challenge, with synthesis capability being the universal limiting factor.

### 4.2.2 KNOWLEDGE DEGRADATION IN SYNTHESIS

Even when models achieve knowledge sufficiency ($\text{KS}(d) = 1$), they often fail to generate the correct answer. We call this *knowledge degradation*: evidence is present in context, yet models forget, ignore, or misuse it during synthesis. To analyse this effect, we focus on two diagnostics, *Forget* and *Lead-astray*, which reveal two distinct synthesis failures: models either fail to recall known knowledge (Forget) or become disrupted by noisy search contexts (Lead-astray). Appendix A shows the formal definition.

**Knowledge degradation patterns.** From Table 2, models with lower Forget and Lead-astray rates consistently achieve stronger Knowledge Utilization, which in turn drives higher Generation Score and Pass@1. High performers such as Grok-4 (Forget 23.65%, Lead-astray 27.70%) and o3-Pro (Forget 21.83%, Lead-astray 25.35%) maintain the best Knowledge Utilization (56.19% and 49.40%), translating into the top Generation Scores (34.71% and 20.86%) and competitive Pass@1 (50.5% and 56.0%). GPT-5 shows a similar profile with very low Forget (17.72%) and strong Knowledge Utilization (49.58%). In contrast, models with severe knowledge degradation see downstream metrics collapse. GPT-OSS-120B, with Forget at 100%, attains only 10.73% Knowledge Utilization and 2.75% Generation Score; Qwen3-30B-Thinking (Forget 79.65%) similarly drops to 13.16% Knowledge Utilization and 7.25% Generation. Mid-tier models such as Gemini-2.5-Flash-Think and Tongyi-DeepResearch-30B follow the same pattern. Overall, the results show that stable evidence retention—not raw retrieval—is the key determinant of deep-search reasoning performance.

**Forgetting dominates misdirection.** Averaging across all models, the mean difference $\text{Forget} - \text{Lead-astray}$ is +10.35% points. This gap indicates that, on **WebDetective**, failures after achieving knowledge sufficiency are more often due to *not using* already-available evidence (forgetting during synthesis) than to being *led astray* by spurious context. In other words, the principal bottleneck lies in evidence integration at answer time rather than in resisting distractors.

### 4.2.3 ROBUSTNESS TO TEST-TIME SCALING

To assess the robustness of our benchmark, we examine test-time scaling (TTS) along two axes. First, we scale context length for a strong ReAct model (Claude-Opus-4.1) to test whether larger budgets improve performance. Second, we vary breadth and iteration counts in **EvidenceLoop** to probe whether extensive exploration can exploit **WebDetective**. These analyses test whether

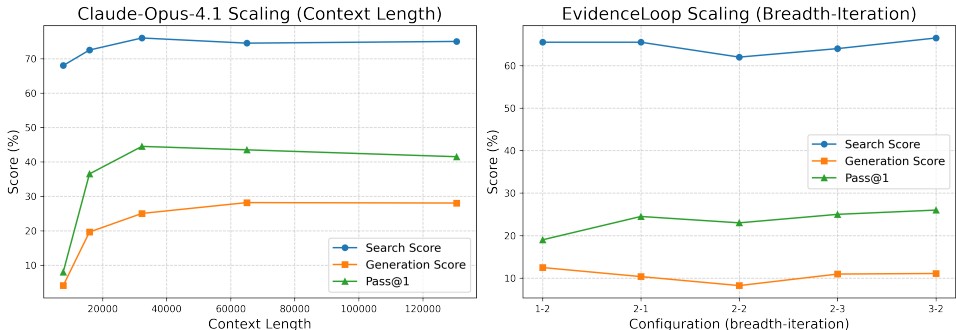

Figure 2: **Test-time scaling (TTS) on WebDetective.** *Left:* Increasing Claude-Opus-4.1's context length boosts Search Score and Pass@1 at shorter ranges, but both plateau beyond 32k tokens, while Generation Score shows only modest gains. *Right:* EvidenceLoop remains stable across breadth–iteration settings, with Pass@1 improving under moderate configurations (e.g., 1–2, 3–2). Generation Score changes little, highlighting synthesis—not search—as the main bottleneck under TTS.

**WebDetective** can be artificially boosted by TTS or instead faithfully reflect underlying system capabilities.

In Figure 2, we observe two main trends. For Claude-Opus-4.1, enlarging the context window from 8K to 32K tokens brings negligible gains: Generation Score plateaus at about 34%, Pass@1 at about 50%, and Search Score increases by less than 1%. For **EvidenceLoop**, expanding the controller from breadth=1, iteration=2 to breadth=3, iteration=2 raises Search Score slightly (45% → 46%, +1%), leaves Generation Score unchanged at 21%, and improves Pass@1 from 49% to 56% (+7%). These results indicate that our benchmark is robust to naïve test-time scaling. Neither larger context budgets nor shallow exploration suffice to "hack" **WebDetective**; achieving further gains requires genuine advances in model reasoning and knowledge utilisation.

## 5 CONCLUSION

We introduce WEBDETECTIVE, a benchmark for evaluating web agents on hint-free multi-hop deep search within a controlled Wikipedia sandbox. Unlike prior datasets that embed reasoning paths (PH) or entity fingerprints (SH), our design enforces autonomous discovery of reasoning chains while enabling fine-grained attribution of failure modes. Crucially, while the sandbox eliminates corpus-based shortcuts, it accommodates multiple valid reasoning strategies—agents may combine partial search with parametric knowledge, and our metrics credit any path achieving knowledge sufficiency. Evaluation of 25 state-of-the-art models reveals consistent weaknesses: systems often retrieve sufficient evidence but fail to utilise it effectively, and appropriate refusals remain nearly absent. Our proposed agentic workflow **EvidenceLoop**, designed to target the specific failure modes our diagnostics reveal, demonstrates that explicit verification and systematic evidence tracking can partially close this gap—improving Knowledge Utilization by 53% over its base model—underscoring that performance cannot be trivially improved by test-time scaling alone. The co-design methodology generalises beyond Wikipedia to any domain with factual content, entity graphs, and masking capability, establishing WebDetective as a framework for developing genuinely autonomous reasoning systems.

## ETHICS STATEMENT

We have read and will adhere to the ICLR Code of Ethics and the ICLR Code of Conduct. Our research introduces WebDetective, a framework for hint-free multi-hop question answering and evidenceLoop, an agentic workflow baseline. The methods used in our study are well-established for academic research. These environments do not contain any personally identifiable information (PII) or sensitive real-world data. Our work did not involve human subjects, crowd-sourcing, or the scraping of private data; therefore, Institutional Review Board (IRB) approval was not required.

We acknowledge that research on autonomous agents carries potential dual-use risks. To mitigate these, our experiments are intentionally confined to benign, closed-world tasks such as online shopping and household activities within simulated settings. We followed good scholarly practice by reporting our methods and results transparently and citing prior work accurately. The authors declare no competing interests or external sponsorships that could have influenced the outcomes of this research.

## REPRODUCIBILITY STATEMENT

We are committed to ensuring the reproducibility of our research. All essential details for reproducing our results are provided within this paper. The WebDetective benchmark design, including the hint-free question formulation principles, the co-designed evaluation system with selective entity masking, and the two-level diagnostic evaluation framework, is thoroughly detailed in the methodology sections. The complete WebDetective dataset statistics, question-environment co-design methodology, and human validation procedures are comprehensively described in the dataset construction sections. Our experimental setup, including the specific language models evaluated (GPT-5, O3-Pro, Claude-Opus-4.1, Gemini-2.5-Pro, Grok-4, Qwen3-235B-Thinking, and others), the controlled Wikipedia sandbox configuration, knowledge sufficiency probing methodology, and evaluation protocols, is fully documented in the experimental sections. The diagnostic metrics formalization, including Knowledge Score, Generation Score, Good Refusal (GR), Knowledge Utilisation (KU), and knowledge degradation tests (Forget and Lead-astray) are rigorously defined in the evaluation framework sections. To facilitate full replication of our benchmark construction pipeline and agent evaluation experiments, we will release our complete codebase, the controlled Wikipedia sandbox environment, a hint-free question dataset with evidence chains, and evaluation scripts as supplementary material.

## ACKNOWLEDGMENT

This research/project is supported by the National Research Foundation, Singapore under its AI Singapore Programme (AISG Award No: AISG3-GV2023-010). This work is also supported by the National Research Foundation, Singapore, under its National Large Language Models Funding Initiative (AISG Award No: AISG-NMLP-2024-005), and NTU SUG project #025628-00001: Post-training to Improve Embodied AI Agents.

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

# Appendix

## Table of Contents

# A  FORMAL METRICS DEFINITION

Traditional multi-hop QA evaluation reduces agent performance to a single pass rate, obscuring the diverse failure modes that occur in complex reasoning tasks. An agent that searches exhaustively but fails to synthesise evidence exhibits fundamentally different limitations than one that refuses prematurely or hallucinates from parametric knowledge. Our co-designed sandbox, with its guaranteed unique reasoning paths, enables unprecedented diagnostic precision in distinguishing these failure modes.

We introduce a two-level evaluation framework that separates *knowledge sufficiency* from *generation quality*. First, we assess whether an agent possesses the requisite knowledge—either through successful search or parametric memory—to answer the question. Second, conditioned on knowledge sufficiency, we evaluate the agent's ability to either correctly synthesise an answer or appropriately refuse when evidence is insufficient.

**Knowledge Sufficiency Assessment:** We assess whether an agent possesses—either through search or parametric knowledge—all information necessary to answer the question. Given the required evidence chain $\mathcal{E} = \{e_1, ..., e_n\}$, we first identify which evidence the agent discovered through search by tracking visited pages in our sandbox. For any missing evidence $e_i \notin \mathcal{E}_{found}$, we then test whether the agent can access this information parametrically.

Specifically, for each missing piece of evidence $e_i$, we construct a focused probe query $p_i$ that tests for that specific knowledge. For instance, if the agent never visited Kane's page and thus missed discovering that "Kane Cornes has brother Chad Cornes," we probe with: "Kane Cornes has brother ______". We define $\text{Probe}(p_i)$ as a function that submits probe $p_i$ to the base model and returns whether the model's response matches the expected answer for evidence $e_i$.

For instance $d$ with evidence chain of length $n_d$, we define:

$$k_i^d = \begin{cases} 1 & \text{if } e_i \in \mathcal{E}_{found} \text{ (found via search)} \\ 1 & \text{if } e_i \notin \mathcal{E}_{found} \wedge \text{Probe}(p_i) = \text{correct} \\ 0 & \text{otherwise} \end{cases}$$

The instance is knowledge sufficient if: $\text{KS}(d) = \prod_{i=1}^{n_d} k_i^d = 1$

We define the overall **Knowledge Score** as the fraction of instances where the agent achieves knowledge sufficiency:

$$\text{KnowledgeScore} = \frac{|\mathcal{S}|}{N} \tag{1}$$

This metric directly measures search effectiveness—a low KnowledgeScore indicates the agent fails to discover necessary evidence through exploration, regardless of its ability to synthesize answers.

**Search Score:** While our masking mechanism enforces the canonical reasoning path $v_0 \rightarrow v_1 \rightarrow ... \rightarrow v_n$, we observe that models may leverage alternative valid reasoning strategies. Specifically, if an entity $v_x$ (reachable through search from $v_0$) has a meaningful relationship to the answer $v_n$ stored in the model's parametric knowledge, the model can combine partial search with memory to reach the correct answer. This represents a legitimate form of reasoning that demonstrates efficient use of both search and parametric knowledge.

To capture this capability, we define **SearchScore** that credits models for finding correct answers through any valid combination of search and parametric knowledge, provided their search efficiency meets or exceeds the ground truth path:

$$\text{SearchScore} = \text{KnowledgeScore} + \frac{|\mathcal{C}|}{N} \tag{2}$$

where $\mathcal{C} = \{d \in \mathcal{D} : \text{correct}(d) \wedge \text{searched}(d) \wedge \text{hops}(d) \leq \text{hops}_{\text{GT}}(d) \wedge \text{KS}(d) = 0\}$ represents instances where the model:

- Produces the correct answer despite not having complete knowledge sufficiency through the canonical path

- Actually performs web search (not relying solely on parametric knowledge)
- Uses no more search hops than the ground truth path length
- Effectively combines discovered entities with parametric knowledge

The requirement that searched$(d)$ = true ensures we only reward genuine search-memory combination strategies, not pure parametric recall. This metric recognizes that effective multi-hop reasoning isn't solely about following predetermined paths, but about efficiently discovering and leveraging available information through intelligent combination of partial search with existing knowledge. The hop constraint ensures models aren't simply performing exhaustive search, but are discovering meaningful connections that enable efficient reasoning.

**Generation Quality Assesement:** Given the knowledge sufficiency assessment, we evaluate generation quality through a conditional framework that captures the fundamental tension in multi-hop QA: agents must synthesize answers when they have sufficient evidence while appropriately refusing when they don't.

Let $\mathcal{D} = \{d_1, ..., d_N\}$ denote the evaluation dataset with $N$ instances. We partition $\mathcal{D}$ along two dimensions:

**Knowledge dimension:**

$$\mathcal{S} = \{d \in \mathcal{D} : \text{KS}(d) = 1\} \quad \text{(knowledge sufficient instances)} \tag{3}$$
$$\mathcal{I} = \mathcal{D} \setminus \mathcal{S} \quad \text{(knowledge insufficient instances)} \tag{4}$$

**Response dimension:**

$$\mathcal{A} = \{d \in \mathcal{D} : \text{agent attempts answer}\} \tag{5}$$
$$\mathcal{N} = \{d \in \mathcal{D} : \text{agent refuses}\} \tag{6}$$

where attempts are further partitioned into $\mathcal{A} = \mathcal{A}_c \cup \mathcal{A}_w$, with $\mathcal{A}_c$ denoting correct answers (matching ground truth) and $\mathcal{A}_w$ denoting wrong answers. Note that $\mathcal{A} \cup \mathcal{N} = \mathcal{D}$.

The intersection of these dimensions creates critical regions that reveal different agent capabilities and failure modes:

- Knowledge sufficient, answers correctly ($\mathcal{S} \cap \mathcal{A}_c$): The ideal scenario—the agent possesses all evidence and successfully synthesizes the correct answer. This demonstrates *knowledge utilization*, the ability to compose multi-hop reasoning without forgetting intermediate steps or being disrupted by irrelevant information.
- Knowledge sufficient, answers wrongly ($\mathcal{S} \cap \mathcal{A}_w$): A synthesis failure—despite having all necessary evidence, the agent produces an incorrect answer. This reveals breakdowns in reasoning composition, where evidence possession doesn't translate to correct synthesis.
- Knowledge sufficient, refuses ($\mathcal{S} \cap \mathcal{N}$): Over-caution—the agent has sufficient evidence but refuses to answer. This represents failure to recognize that the evidence chain is complete, missing opportunities to provide helpful answers.
- Knowledge insufficient, refuses ($\mathcal{I} \cap \mathcal{N}$): The second ideal scenario—*good refusal*. The agent lacks critical evidence and appropriately declines to answer, demonstrating epistemic awareness and avoiding hallucination.
- Knowledge insufficient, attempts answer ($\mathcal{I} \cap \mathcal{A}$): The most problematic behavior—the agent lacks evidence but attempts an answer anyway (whether correct by luck or wrong), typically through hallucination, guessing, or over-reliance on partial information.

This visualization reveals that generation quality isn't monolithic—an agent might excel at refusing when uncertain but fail to synthesize known information, or vice versa. For instance, an overly conservative agent might achieve perfect good refusal by refusing everything (large $\mathcal{N}$ region), while an overly confident agent might attempt every question (large $\mathcal{A}$ region) leading to frequent hallucinations in the $\mathcal{I}$ zone.

To capture these complementary capabilities, we define two core metrics:

**Good Refusal (GR)** measures the agent's ability to appropriately abstain when lacking evidence. It evaluates $\mathcal{N}$'s overlap with $\mathcal{I}$—high recall indicates the agent successfully avoids hallucination by refusing most knowledge-insufficient cases, while high precision ensures refusals are justified (not bleeding unnecessarily into $\mathcal{S}$).

$$\text{Recall}_{\text{GR}} = \frac{|\mathcal{N} \cap \mathcal{I}|}{|\mathcal{I}|}, \quad \text{Precision}_{\text{GR}} = \frac{|\mathcal{N} \cap \mathcal{I}|}{|\mathcal{N}|}, \quad \text{F1}_{\text{GR}} = 2 \cdot \frac{\text{Recall}_{\text{GR}} \cdot \text{Precision}_{\text{GR}}}{\text{Recall}_{\text{GR}} + \text{Precision}_{\text{GR}}} \quad (7)$$

**Knowledge Utilization (KU)** assesses the agent's ability to synthesize correct answers when evidence is available. It examines $\mathcal{A}_c$ within $\mathcal{S}$—high recall means the agent leverages available evidence effectively, while high precision indicates that attempts are typically grounded in sufficient knowledge.

$$\text{Recall}_{\text{KU}} = \frac{|\mathcal{A}_c \cap \mathcal{S}|}{|\mathcal{S}|}, \quad \text{Precision}_{\text{KU}} = \frac{|\mathcal{A}_c \cap \mathcal{S}|}{|\mathcal{A}|}, \quad \text{F1}_{\text{KU}} = 2 \cdot \frac{\text{Recall}_{\text{KU}} \cdot \text{Precision}_{\text{KU}}}{\text{Recall}_{\text{KU}} + \text{Precision}_{\text{KU}}} \quad (8)$$

Importantly, these metrics are non-competing—improving one shouldn't decrease the other in a well-designed system. An ideal agent achieves high $\text{F1}_{\text{GR}}$ (refusing when and only when knowledge is insufficient) while maintaining high $\text{F1}_{\text{KU}}$ (correctly answering when evidence is available). To capture both capabilities while preventing gaming, we define a unified **Generation Score**:

$$\text{GenScore} = \frac{\text{F1}_{\text{GR}} + \text{F1}_{\text{KU}}}{2} \cdot \frac{|\mathcal{S}|}{N} \quad (9)$$

The $|\mathcal{S}|/N$ weighting (KnowledgeScore) is crucial for preventing metric exploitation: without it, models could game the evaluation by adopting a degenerate strategy—performing minimal search and refusing nearly all questions. Such a model would achieve high $\text{F1}_{\text{GR}}$ (correctly refusing the many knowledge-insufficient cases) while contributing nothing useful, yet still obtain a substantial GenScore. This creates a perverse incentive where models might optimize for conservative refusal rather than improving search capabilities. The weighting ensures that models cannot exploit the evaluation structure—they must demonstrate effective evidence discovery to achieve competitive scores, aligning the metric incentives with the actual goal of multi-hop reasoning systems.

Unlike simple pass rates, our metrics provide actionable insights: low KnowledgeScore indicates inadequate search strategies, low GR scores reveal over-confident hallucination, and low KU scores expose synthesis failures despite having evidence. This diagnostic precision, enabled by our co-designed evaluation environment, illuminates the specific capabilities required for robust multi-hop reasoning.

**Knowledge Forget Test.** We test $\text{LM}(q, \mathcal{E}_{\text{found}})$ where $\mathcal{E}_{\text{found}} = \mathcal{E}_{\text{visited}} \cap \mathcal{E}_{\text{GT}}$ represents evidence from ground-truth URLs that the model actually visited. When this fails despite $\text{KS}(d) = 1$, it reveals *knowledge forget*: the model cannot leverage its parametric knowledge to fill missing pieces when answering the full question, even though it correctly answers individual probes $\text{Probe}(p_i)$ for each missing evidence $e_i \in \mathcal{E}_{\text{GT}} \setminus \mathcal{E}_{\text{found}}$.

**Lead-astray Test.** When $\text{LM}(q, \mathcal{E}_{\text{found}})$ succeeds but the model fails in its actual search trajectory, we identify *lead-astray*: the model can synthesize the answer from clean evidence but is disrupted by the accumulated search context (failed attempts, irrelevant pages, exploration noise).

Formally, for the set of knowledge-sufficient instances $\mathcal{S}^* = \{d \in \mathcal{D} : \text{KS}(d) = 1 \wedge \text{incorrect}(d)\}$ where the model fails despite having all necessary knowledge:

$$\text{ForgetRate} = \frac{|\{d \in \mathcal{S}^* : \text{LM}(q_d, \mathcal{E}_{\text{found}}^d) \neq a_d^*\}|}{|\mathcal{S}^*|}$$

$$\text{LeadAstrayRate} = \frac{|\{d \in \mathcal{S}^* : \text{LM}(q_d, \mathcal{E}_{\text{found}}^d) = a_d^* \wedge \text{actual\_output}(d) \neq a_d^*\}|}{|\mathcal{S}^*|}$$

These metrics decompose knowledge-sufficient failures: ForgetRate identifies when models cannot integrate parametric knowledge with partial search results, while LeadAstrayRate reveals when noisy search trajectories corrupt otherwise successful reasoning.

## B DATASET CONSTRUCTION

To instantiate our hint-free multi-hop QA benchmark, we develop a systematic pipeline that transforms single-hop Wikipedia QA pairs into verified multi-hop reasoning chains while ensuring each hop is necessary for answering.

**Source Data and Chain Discovery.** We begin with a corpus of Wikipedia-based QA pairs where each question targets a specific paragraph on a Wikipedia page (the starting entity $v_0$) and has an answer that is another Wikipedia entity ($v_n$). These seed questions are designed to be unambiguous and simple, avoiding any linguistic hints about reasoning paths. To construct multi-hop chains, we first block the direct connection between $v_0$ and $v_n$, then perform breadth-first search (BFS) to find the shortest alternative path $v_0 \rightarrow v_1 \rightarrow ... \rightarrow v_n$ through Wikipedia's hyperlink graph. For each edge $(v_i, v_{i+1})$ in the discovered path, we extract the sentence $e_i$ from $v_i$'s Wikipedia page that contains the hyperlink to $v_{i+1}$, forming the evidence chain $\mathcal{E} = \{e_1, e_2, ..., e_n\}$.

**Verification of Reasoning Necessity.** Not all discovered paths constitute valid answers to the question—most arbitrary paths from $v_0$ to $v_n$ through Wikipedia's link graph are completely unrelated to what the question asks. For instance, a path connecting two people through their universities and shared colleagues is irrelevant for a question asking about family relationships. We implement a three-stage verification process using a strong language model (Qwen-3-235B in our implementation), denoted as $\text{LM}(\cdot)$ which takes text input and generates an answer:

1. **Parametric Inaccessibility**: We verify that $\text{LM}(q) \neq v_n$, ensuring the answer cannot be directly retrieved from the model's parametric memory without evidence.

2. **Evidence Sufficiency**: We confirm that $\text{LM}(q, \mathcal{E}) = v_n$, validating that the complete evidence chain enables correct answer generation.

3. **Evidence Necessity**: For each evidence piece $e_i$, we verify that $\text{LM}(q, \mathcal{E} \setminus \{e_i\}) \neq v_n$, ensuring every hop in the chain is required for reasoning. This ablation test eliminates questions where evidence pieces are redundant or where shortcuts exist.

**Human Validation and Dataset Statistics.** Questions passing automated verification undergo human annotation by 2 researchers with NLP expertise. Each question is independently reviewed following a structured protocol:

1. **Annotation Protocol**: For each question, annotators receive the question $q$, evidence chain $\mathcal{E} = \{e_1, ..., e_n\}$, and answer $v_n$. They verify three criteria:
   - *Necessity*: Whether the question can be answered without the evidence chain using only general knowledge
   - *Sufficiency*: Whether the evidence chain logically derives the answer without requiring external information
   - *No hints*: Whether the question avoids linguistic cues that reveal intermediate reasoning steps

2. **Validation Process**: Each question requires 2-3 minutes of review. Annotators trace through the reasoning chain step-by-step, attempting to answer the question both with and without the evidence to ensure all pieces are necessary. Questions where intermediate entities could be guessed from the question phrasing or where the evidence chain has logical gaps are rejected.

3. **Dataset Filtering**: Of approximately 450 machine-verified questions reviewed, 200 questions ($\sim$44%) pass human validation. Common rejection reasons include: evidence chains not targeting the questions, evidence chains with missing logical connections, and questions containing subtle hints about the reasoning path (e.g., mentioning attributes that implicitly identify intermediate entities).

This manual verification process, totaling approximately 20 hours of annotation effort, ensures our final dataset contains only questions that genuinely require discovering and composing the complete multi-hop reasoning chain.

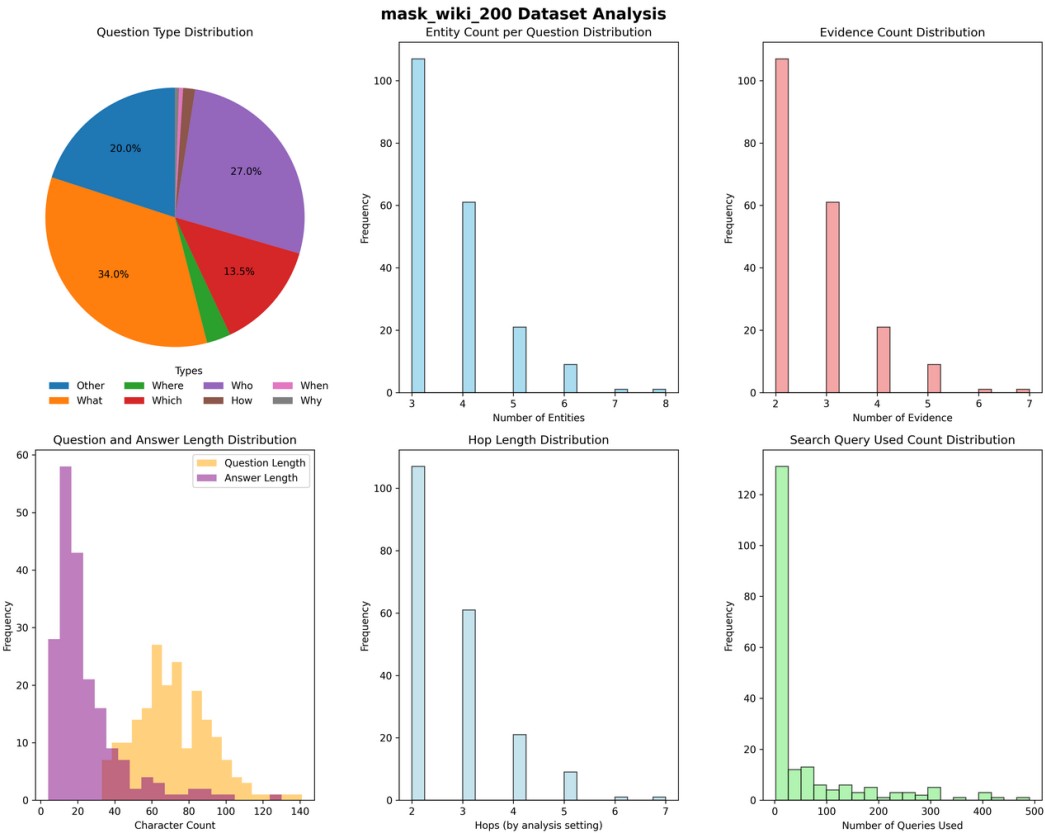

Figure 3: Dataset statistics for WebDetective benchmark. The figure shows: (a) Distribution of question types, (b) Number of entities per question, (c) Evidence count distribution, (d) Question and answer length in characters, (e) Hop length distribution by analysis setting, and (f) Search query usage patterns. The dataset exhibits controlled complexity with predominantly 2-3 hop questions while maintaining challenging longer chains.

## C   DATASET STATISTICS

Our final WebDetective benchmark comprises 200 hint-free multi-hop questions, carefully curated through our verification pipeline. Figure 3 presents a comprehensive analysis of the dataset characteristics.

**Question Complexity.** The dataset exhibits controlled complexity suitable for diagnostic evaluation. Questions require 2 to 6 hops of reasoning (mean: 2.85 hops), with the distribution heavily weighted toward 2-hop (55%) and 3-hop (31%) questions, while maintaining a challenging subset of 4+ hop questions (14%). This distribution balances tractability with sufficient complexity to stress-test multi-hop reasoning capabilities. Each question involves 3 to 8 Wikipedia entities (mean: 3.73), with the modal question requiring exactly 3 entities to form the complete reasoning chain.

**Question Types and Domains.** The dataset spans diverse question types, with "What" questions comprising 34% of the dataset, "Who" questions 27%, and "Which" questions 13.5%, ensuring broad coverage of information-seeking patterns. Questions are concise (mean: 71.4 characters) with typically short answers (mean: 28.6 characters), reflecting natural information needs without verbose specifications that might hint at reasoning paths.

**Evidence Requirements.** The evidence distribution aligns with hop counts, with most questions requiring 2-3 pieces of evidence (52.5% and 31% respectively). This controlled evidence requirement

enables precise diagnosis of where reasoning fails—whether at initial discovery, intermediate steps, or final synthesis.

The dataset's careful balance of complexity, diversity, and diagnostic precision makes it suitable for evaluating the full spectrum of multi-hop reasoning capabilities, from basic 2-hop familial relationships to complex 5-hop chains requiring sustained context retention across multiple search iterations.

**Domain Coverage.** The dataset spans diverse topical domains to ensure broad evaluation coverage, as shown in Table 4. This distribution ensures WebDetective is not skewed toward any single relation type (such as familial ties) but instead covers reasoning patterns essential for comprehensive evaluation.

Table 4: Domain distribution in the WebDetective benchmark.

| Domain | Percentage |
|---|---|
| Film, Television & Theatre | 20.0% |
| History, Politics & Military | 18.0% |
| Geography, Locations & Landmarks | 16.0% |
| Music & Audio Production | 13.5% |
| Sports & Competitions | 11.5% |
| Biography, Genealogy & Education | 8.5% |
| Literature, Comics & Pop Culture | 6.5% |
| Science, Technology & Transport | 6.0% |

## D    STATISTICAL CONSIDERATIONS

**Quality-Focused Construction Pipeline.** The relatively modest dataset size (200 questions) is a direct consequence of our strict construction and filtering pipeline. For each candidate question-answer pair $(q, a^*)$ with evidence chain $\mathcal{E} = \{e_1, ..., e_n\}$, we enforce three conditions: (1) Parametric Inaccessibility: the verifier LM cannot produce $a^*$ from $q$ alone; (2) Evidence Sufficiency: $\text{LM}(q, \mathcal{E}) = a^*$; and (3) Evidence Necessity: for every $i \in \{1, ..., n\}$, we probe with the incomplete set $(q, \mathcal{E} \setminus \{e_i\})$ and verify failure. The majority of automatically generated candidates are discarded during this rigorous process, followed by manual verification.

**Confidence Intervals.** Our binomial confidence analysis shows that with $n = 200$, we achieve a 95% confidence interval of $\pm 6.9\%$ (assuming 50% pass rate). While this is sufficient for identifying the large-scale failure patterns our benchmark targets—the gap between Knowledge Score (79%) and Generation Score (23%) for top models, the near-universal failure of refusal capabilities ($< 30\%$ Good Refusal F1), and the synthesis bottleneck (Knowledge Utilization peaks at 56%)—these are substantial effects that do not require larger samples to detect reliably.

**Diagnostic Purpose.** Our primary aim with WebDetective is to provide a diagnostic benchmark that exposes fundamental failure modes rather than ranking systems with marginal differences. The patterns we identify are consistent across all 25 evaluated systems, demonstrating that even with our 2-3 hop concentration (86% of questions), the benchmark remains far from saturated: the highest Pass@1 is only 56% (o3-Pro), and state-of-the-art agents frequently fail to synthesize 2-3 pieces of evidence without hints.

## E    THE EVIDENCELOOP FRAMEWORK

The hint-free nature of our benchmark exposes fundamental limitations in current multi-hop reasoning approaches. Without linguistic scaffolding, agents must autonomously discover which connections matter among thousands of facts—a challenge that, as our results show, causes even state-of-the-art models to achieve only  50% accuracy. To better understand these challenges and establish a baseline for future work, we design an agentic workflow that explicitly targets the unique difficulties our benchmark reveals: the need for broad exploration without context explosion, evidence retention across long trajectories, and synthesis from accumulated but noisy search contexts.

## E.1 CORE ARCHITECTURE: ITERATIVE REFINEMENT WITH FALLBACK

Our framework attempts to balance exploration breadth with computational feasibility through $R_{\max}$ iterations. Illustrated in Figure 4, each iteration $r$ launches $N$ parallel solver agents $\{A_1^r, ..., A_N^r\}$ that explore different reasoning paths simultaneously. Each agent $A_i^r$ receives the question $q$, an aggregated context $C^r$ from previous iterations (with $C^0 = \emptyset$ initially), and executes up to $B$ actions.

After each iteration, we employ a two-stage refinement process:

1. An **extraction agent** processes the reasoning contexts from all $N$ parallel agents to identify key findings, evidence references, and promising paths

2. An **aggregation agent** synthesizes these extracted insights into a refined context $C^{r+1}$ for the next round, preserving valuable discoveries while discarding exploration noise

This iterative refinement addresses a core challenge our benchmark exposes: early rounds might explore many directions—sports connections, geographic locations, family relations—but the extraction-aggregation pipeline identifies which paths warrant deeper exploration, preventing the context explosion that causes single-pass approaches to fail while avoiding premature path commitment. If no conclusive answer emerges after $R_{\max}$ iterations, a final aggregation agent consolidates all discovered evidence into a comprehensive context $C^{\text{final}}$. This context is then passed to a synthesis-only solver that attempts to derive the answer purely from the accumulated evidence without additional search actions—effectively testing whether the failure stems from insufficient exploration or poor evidence composition.

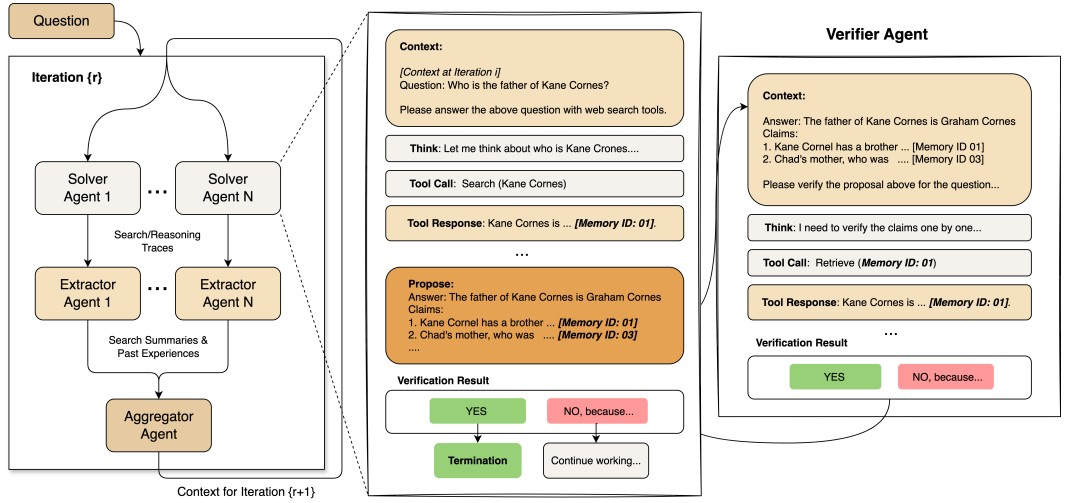

Figure 4: Overview of the **EvidenceLoop** framework. The system employs parallel solver and extractor agents that perform search and reasoning to generate proposals with supporting claims. These proposals are then verified through memory retrieval, with the aggregated context fed into subsequent iterations until verification succeeds or a termination condition is met.

## E.2 EVIDENCE MEMORY SYSTEM

Enabling this iterative refinement is our Evidence Memory System $\mathcal{M}$. When any agent performs a search or visits a page, the system: 1) Stores complete results in persistent memory; 2) Assigns unique Evidence IDs (EIDs) for reference; and 3) Returns both full content and EID to the agent.

The EID system serves multiple critical functions in our framework. First, during extraction and aggregation between iterations, the extraction and aggregation agent produces summaries that preserve EID references alongside extracted facts—for example, "Kane has brother Chad [EID-042], Chad's stepmother is Nicole [EID-089]". This allows subsequent solver agents to receive concise, actionable summaries while retaining the ability to retrieve full evidence on demand through the

retrieve action as an external tool, which takes an EID and returns the complete original content from memory. Second, these EIDs enable systematic verification (detailed in Appendix E.3), where verification agents can trace claims back to their original sources and validate reasoning chains against the actual evidence.

The memory system transforms how evidence flows through iterations. Rather than forcing agents to work with either overwhelming full documents or lossy compressions, agents can work with focused summaries while maintaining access to complete evidence through EID-based retrieval. This design ensures that even as contexts become more refined across iterations, agents never lose access to the complete evidence trail that supports their reasoning, allowing them to dive deep into specific evidence when needed for detailed analysis or verification.

### E.3 VERIFICATION: ENSURING EVIDENCE-GROUNDED REASONING

The verification mechanism prevents premature or hallucinated answers from propagating through our system. When any solver agent $A_i^r$ proposes an answer, it must decompose the answer into atomic claims $\{c_1, c_2, ..., c_m\}$, where each claim $c_j$ is explicitly linked to an EID from the memory system—e.g., "Kane has brother Chad [EID-042]". No unsupported claims are permitted.

The verification agent $V$ evaluates each proposal:

$$V(q, \text{answer}, \{c_j, \text{EID}_j\}_{j=1}^m) \to \{\text{YES}, \text{NO(feedback)}\}$$

For each claim-evidence pair, the verifier retrieves the full content from $\mathcal{M}$ via the EID and validates: (1) whether the source genuinely entails the claimed fact, (2) whether the claims collectively derive the answer, and (3) whether the answer correctly addresses the original question.

Verification occurs *during* solver execution. Rejections provide specific feedback back to the solver, allowing immediate gap-filling within the remaining action budget $B$, while acceptance immediately terminates all iterations. This ensures both evidence grounding and computational efficiency—solvers can correct incomplete reasoning in real-time while avoiding unnecessary exploration once the answer is verified.

### E.4 ABLATIONS ON ARCHITECTURE DESIGN

To quantify the contributions of the memory and verification components, we conduct ablation studies using `DeepSeek-R1-0528` as the base model. As shown in Table 5, memory-driven working-context structuring is the main force behind alleviating knowledge-related deficiencies, while verification plays a complementary role by strengthening precision and ensuring robust utilization of retrieved knowledge.

**Impact of Memory.** Removing the structured evidence buffer (*w/o Memory*) leads to the most substantial performance drop, with the Search Score decreasing by **8.5%** (71.5% → 63.0%). This underscores that memory-driven working-context structuring is the primary contributor to mitigating knowledge-related shortages and enabling effective knowledge discovery.

**Impact of Verification.** Eliminating the explicit verification loop (*w/o Verification*) results in a moderate but consistent reduction in both Search Score (−2.9%) and Knowledge Utilization F1 (−1.54%). This indicates that verification provides valuable additional support, improving retrieval quality and enhancing the stability of information synthesis.

Table 5: Ablation study on **EvidenceLoop**. Search Score and Knowledge Utilization F1 are reported, with emphasis on the memory and verification modules, identified earlier as key failure points.

| Model Variant | Search Score (%) | Knowledge Util. F1 (%) |
|---|---|---|
| **EvidenceLoop** (DeepSeek-R1-0528) | 71.50 | 25.00 |
| − w/o Memory | 63.00 | 24.38 |
| − w/o Verification | 68.60 | 23.46 |

Table 6: Performance comparison across different base models. **EvidenceLoop** consistently outperforms the standard baselines on `DeepSeek-R1-0528` and `GLM-4.5-Air`, particularly in Knowledge Utilization and overall Pass@1 accuracy. Relative improvements are indicated in parentheses.

| Model & Method | Knol. Suff | Search Sc. | Knol. Util F1 | Pass@1 |
|---|---|---|---|---|
| *Backbone: DeepSeek-R1-0528* | | | | |
| Baseline | 61.0 | 65.5 | 16.36 | 20.5 |
| **EvidenceLoop** | **68.0** | **71.5** | **25.00** | **28.5** |
| Δ | *+7.0* | *+6.0* | *+53% (rel.)* | *+39% (rel.)* |
| *Backbone: GLM-4.5-Air* | | | | |
| Baseline | 55.5 | 60.5 | 17.97 | 19.0 |
| **EvidenceLoop** | **62.0** | **63.5** | **19.70** | **23.5** |
| Δ | *+6.5* | *+3.0* | *+10% (rel.)* | *+24% (rel.)* |

### E.5 GENERALIZATION ACROSS BASE MODELS

To assess whether the benefits of **EvidenceLoop** rely on specific model characteristics or generalize across different backbones, we extended our evaluation to two distinct architectures: `DeepSeek-R1-0528` and `GLM-4.5-Air`. This analysis specifically tests the framework's robustness against context degradation and limited search coverage inherent in standard ReAct-style baselines.

As detailed in Table 6, **EvidenceLoop** delivers consistent improvements in knowledge discovery and synthesis across both architectures. Specifically, we observe:

- **Search Coverage:** Search Scores improve by +3.0 to +6.0 points, indicating a more effective exploration of the information space.

- **Knowledge Utilization:** The structured evidence buffer and verification loop yield substantial gains in utilization, with relative improvements in F1 Score ranging from $10\%$ to $53\%$.

- **Task Performance:** These gains translate directly to downstream accuracy, boosting Pass@1 by +4.5 to +8.0 points absolute (+24% to +39% relative).

Most notably, the `DeepSeek-R1-0528` backbone achieves a $53\%$ relative improvement in Knowledge Utilization F1 ($16.36\% \rightarrow 25.0\%$). This result confirms that **EvidenceLoop** effectively mitigates the synthesis bottleneck identified by **WebDetective**, transforming retrieved raw context into actionable evidence through its iterative explore-verify mechanism.

## F FAILURE CASE STUDIES

We identify four recurring failure patterns through qualitative analysis:

- **Premature Search Termination:** After a small amount of failed searches, models enter a "learned helplessness" state where they immediately conclude the answer is unavailable and refuse further exploration. Even explicit prompting like "please continue searching" fails to restart the search process, with models insisting no more information exists in the sources. This occurs despite obvious next steps being available—for instance, finding Chad as Kane's brother but refusing to visit Chad's page directly, instead declaring the search futile.

- **Context-Induced Instruction Degradation:** As search context accumulates, models progressively lose their ability to follow basic instructions. In shorter context, they proper use proper `<answer>` tags and structured reasoning, but gradually degrade to dropping tags intermittently, then completely abandoning format requirements, producing stream-of-consciousness text without capitalization or proper syntax.

- **Evidence Tracking Failure:** Models lose track of discoveries across search iterations, repeatedly searching for entities they've already found or failing to maintain entity relationships discovered earlier. They cannot distinguish between "not found due to masking" versus "haven't searched yet," leading to redundant searches or premature abandonment of viable paths. The accumulated context becomes noise rather than useful evidence.
- **Redundant Search Loops:** Models frequently re-visit or re-search pages they've already explored, especially after intermediate reasoning steps. For example, after visiting Kane's page, finding Chad, visiting Chad's page, then spending time reasoning about the relationships, the model might search for "Kane Cornes" again or revisit Kane's Wikipedia page, essentially restarting from scratch. While not technically incorrect, this pattern wastes action budget and rapidly fills context with duplicate information, accelerating context degradation and reducing the effective search depth the model can achieve before hitting token limits or action budgets.

## G  RELATED WORK

### G.1  MULTI-HOP QUESTION ANSWERING BENCHMARKS

Multi-hop QA benchmarks evaluate models' ability to compose information across multiple reasoning steps. Early datasets like HotpotQA Yang et al. (2018b) and WikiHop Welbl et al. (2018) established foundational evaluation frameworks but suffer from systematic biases. Recent benchmarks have expanded coverage: FanOutQA Zhu et al. (2024) addresses multi-document reasoning, MINTQA He et al. (2024) targets long-tail knowledge with 28K+ questions, and MEQA Li et al. (2024b) focuses on event-centric reasoning chains. However, these benchmarks embed hints that fundamentally alter the reasoning task.

We identify two categories of hints prevalent in existing benchmarks. **Path-hinting** occurs when questions linguistically encode reasoning chains (e.g., "What dance academy did the starring actress from The Glory of Tang Dynasty graduate from?"), reducing the task to executing pre-specified steps. **Specification-hinting** provides excessive constraints that make answers discoverable through constraint satisfaction rather than reasoning (e.g., combining "East German team," "founded 1966," "player born in 90s"). Unlike MuSiQue Trivedi et al. (2022) or 2WikiMultiHopQA Ho et al. (2020), which contain implicit structural hints, WebDetective introduces genuinely hint-free questions requiring autonomous reasoning path discovery.

### G.2  RETRIEVAL-AUGMENTED GENERATION AND AGENTS

The evolution from static RAG pipelines to agentic architectures represents a fundamental shift in how LLMs interact with external knowledge Singh et al. (2025); Oche et al. (2025). While traditional RAG systems like TRACE Fang et al. (2024) achieve improvements through knowledge-grounded reasoning chains, they operate within predetermined patterns. Agentic RAG systems employ adaptive strategies: Adaptive-RAG Jeong et al. (2024) adjusts retrieval depth based on question complexity, while graph-based approaches like GNN-Ret Li et al. (2024c) and HopRAG Liu et al. (2025) leverage graph neural networks for multi-hop reasoning, achieving 10% accuracy improvements on benchmarks like 2WikiMQA.

Recent advances in 2025 emphasize diverse reasoning paths. DP-CoT Li et al. (2024a) addresses single-path limitations through passage-level and sentence-level evidence generation. However, our evaluation reveals these advances fail to overcome hint-free challenges: median Generation Scores of 20% across tested models indicate current architectures cannot effectively discover reasoning chains without linguistic scaffolding.

### G.3  EVALUATION FRAMEWORKS

Traditional metrics like exact match and F1 scores collapse diverse failure modes into single values, obscuring why models fail Kwiatkowski et al. (2019); Petroni et al. (2021). Recent frameworks attempt more nuanced evaluation: RAGAS Shahul et al. (2024) provides reference-free RAG metrics, while RAGTruth Niu et al. (2024) enables hallucination analysis. For agents, AgentBench Liu et al. (2023) evaluates across eight environments, tau-bench Yao et al. (2024) addresses multi-turn

interactions, and TheAgentCompany Xu et al. (2025) introduces workplace tasks with simulated colleagues.

Web-based benchmarks have evolved significantly. WebArena Zhou et al. (2023) provides realistic web environments requiring long-horizon planning but lacks controlled evaluation for precise failure attribution. SWE-bench Jimenez et al. (2024) evaluates code generation on GitHub issues, with SWE-bench Verified OpenAI (2024) addressing underspecified problems. While these benchmarks test complex capabilities, they don't address the specific challenge of verifying multi-hop reasoning paths.

Our diagnostic framework decomposes evaluation into *knowledge sufficiency* (whether agents possess required evidence) and *conditional generation quality* (synthesis given sufficient knowledge). This separation reveals that models achieve 79% knowledge sufficiency but only 23% generation scores, indicating synthesis and relevance determination—not search—as primary bottlenecks.

## H   LLMs Usage

LLMs were used to polish the writing.

