# OpenReview forum: "Demystifying Deep Search: A Holistic Evaluation with Hint-free Multi-Hop Questions and Factorised Metrics"
_ICLR.cc/2026/Conference — ICLR 2026 Poster_

### Official Review · Reviewer_rUPx · 2025-10-19

**Soundness:** 3
**Presentation:** 3
**Contribution:** 3
**Rating:** 6
**Confidence:** 4

**Summary:**

This paper introduces WebDetective, a new benchmark for evaluating web agents on hint-free multi-hop deep search tasks.
Unlike prior datasets that leak reasoning paths (path-hinting) or entity fingerprints (spec-hinting), the authors construct a controlled Wikipedia sandbox that enforces step-by-step reasoning without shortcuts.
They also propose a two-level evaluation framework that disentangles knowledge sufficiency, utilization, and refusal.
Experiments on 25 state-of-the-art models show systematic weaknesses: models often retrieve enough evidence but fail at synthesis, and almost none refuse appropriately.
The authors further design EvidenceLoop, an agentic workflow baseline that uses verification loops and memory tracking to partially address these gaps.

**Strengths:**

- The paper identifies and argues against the hidden hinting in existing multi-hop deep research datasets.
- The co-design of hint-free questions with a sandbox masking mechanism is clever and ensures true multi-hop reasoning.
- 25 strong models are compared, giving a broad empirical picture.

**Weaknesses:**

- The controlled Wikipedia sandbox may be too artificial compared to real-world open web environments.
- While an interesting method, improvements seem modest. It feels more like a proof-of-concept than a strong new method for EvidenceLoop.

**Questions:**

- Which LLM does your method use in Table 1?

---

> ### Author Response · Authors · 2025-11-24
> **Response to Reviewer rUPx**
>
> Thank you for your thoughtful review and recognition of our contributions, particularly our identification of hidden hinting in existing benchmarks and our co-designed evaluation framework. We address your concerns below:
>
>
> ### **W1: Controlled Wikipedia sandbox vs. real-world open web**
>
> We argue that while the *boundary* of our sandbox is controlled, the *reasoning process* it demands is highly representative of real-world challenges, and the framework itself is designed for broad extensibility.
>
> **1. Realistic Reasoning Constraints**
>
> We utilized Wikipedia primarily as a controlled **initial testbed** to validate our diagnostic metrics. However, unlike traditional academic datasets that provide "gold paragraphs," our **hint-free design** forces the agent to navigate a noise-heavy environment without explicit guidance. This accurately mirrors the core difficulty of real-world open-web search—filtering vast amounts of irrelevant data to find specific evidence—even within a closed corpus.
>
> **2. Extensibility to the Open Web** Crucially, our "co-design" methodology is not limited to Wikipedia. The framework requires only three generic components to function:
>
> 1. **A Text Corpus** (factual content).
>
> 2. **A Link Structure** (entity-level graph).
>
> 3. **Masking Capability** (to enforce reasoning).
>
> Because these requirements are minimal, the same pipeline (path construction + masking + factorized evaluation) can be readily applied to open-web snapshots, news archives, or domain-specific data. Thus, the current benchmark serves as a foundational proof-of-concept for a diagnostic protocol that is intended to scale to unrestricted open-web environments in future work.
>
>
> ### **W2: EvidenceLoop as a meaningful baseline**
>
> We appreciate the reviewer's assessment of EvidenceLoop's improvements. While the absolute gains may appear moderate, they represent meaningful progress when viewed in context. Comparing EvidenceLoop directly to its base model (DeepSeek-R1) reveals notable improvements:
>
> - **Pass@1**: Improves from 20% to 25% (**25% relative improvement**)
> - **Generation Score**: Increases from 10.57 to 12.61 (**19% relative improvement**)
> - **Knowledge Utilization F1**: Improves from 15.55 to 23.79 (**53% relative improvement**)
>
> The **53% improvement in Knowledge Utilization** is particularly noteworthy—it demonstrates that our structured evidence buffer and verification loop directly address the synthesis bottleneck that WebDetective identifies as a critical failure mode across all tested models.
>
> EvidenceLoop's key architectural differences from standard ReAct include:
> 1. **Two-phase controller**: Explicit separation of exploration and verification phases
> 2. **Structured evidence memory**: Maintains (entity, snippet, source) tuples rather than flat dialogue history
> 3. **Verification loop tied to refusal**: Re-derives answers using only the evidence buffer, defaulting to refusal when evidence is incomplete
>
> To further demonstrate that *EvidenceLoop meaningfully improves over ReAct-style methods—particularly under context degradation and limited search coverage*—we additionally evaluate it on two different model architectures (DeepSeek-R1-0528 and GLM-4.5-Air), testing whether the gains generalize beyond a single backbone.
>
> Across both architectures, EvidenceLoop shows *consistent improvements* in knowledge discovery, utilisation, and overall task accuracy, as reflected in the results below.
>
> | Model                        | Knowledge Suff | Search Score  | Knowledge Util F1  | Pass@1             |
> | ---------------------------- | -------------- | ------------- | ------------------ | ------------------ |
> | DeepSeek-R1-0528                  | 61.0           | 65.5          | 16.36              | 20.5               |
> | *EvidenceLoop (DeepSeek-R1-0528)* | *68.0 (+7.0)*  | *71.5 (+6.0)* | *25.0 (+53% rel.)* | *28.5 (+39% rel.)* |
> | GLM-4.5-Air                  | 55.5           | 60.5          | 17.97              | 19.0               |
> | *EvidenceLoop (GLM-4.5-Air)* | *62.0 (+6.5)*  | *63.5 (+3.0)* | *19.7 (+10% rel.)* | *23.5 (+24% rel.)* |
>
> These cross-architecture results confirm that EvidenceLoop’s iterative retrieval loop provides *generalizable gains* beyond standard ReAct.

---

> ### Author Response · Authors · 2025-11-24
> **Response to Reviewer rUPx**
>
> ### **Q1: Which LLM does the method use in Table 1?**
>
> For EvidenceLoop in Table 1, we used **DeepSeek-R1** as the base model. The comparison shows our workflow improves DeepSeek-R1's performance from 20% to 25% Pass@1. We will make this explicit in the paper text rather than relying solely on the table notation.

---

### Official Review · Reviewer_yEz4 · 2025-10-26

**Soundness:** 3
**Presentation:** 3
**Contribution:** 4
**Rating:** 6
**Confidence:** 4

**Summary:**

This paper presents WebDetective, a benchmark for evaluating web agents on hint-free multi-hop deep search tasks within a controlled Wikipedia sandbox. Unlike prior datasets that leak reasoning paths or entity attributes, it enforces autonomous discovery of reasoning chains and enables fine-grained attribution of failure modes. The authors also propose a factorized evaluation framework—separating knowledge sufficiency, generation quality, and refusal behavior—and an agentic baseline (EvidenceLoop) integrating evidence tracking and verification. Experiments on 25 state-of-the-art models show that while systems can retrieve sufficient evidence, they often fail to synthesize it effectively, exposing a major synthesis bottleneck. The benchmark demonstrates that performance gains cannot be achieved by scaling context or computation, emphasizing that true progress in deep reasoning requires better evidence utilization and calibration, rather than brute-force search or larger models.

**Strengths:**

1. The paper proposes a very interesting and promising setting, hint-free multi-hop deep search within a controlled Wikipedia sandbox, which effectively exposes reasoning weaknesses that are usually hidden in traditional benchmarks with embedded hints. The benchmark design is well-motivated and enables fine-grained diagnosis of model behavior.
2. The experiments are comprehensive, covering 25 state-of-the-art systems across multiple providers, and the analysis is insightful, identifying key failure modes like synthesis bottlenecks and calibration issues.
3. The additional analyses are also strong, including detailed model profiling and robustness studies on scaling context length and computational budgets, which further validate the benchmark’s diagnostic value.

**Weaknesses:**

1. The presentation could be improved, including writing clarity, figure design, and overall structure. Some key insights are buried in the text and would benefit from clearer highlighting and visual emphasis.
2. Certain methodological details are missing or underexplained. For instance, the process for testing Parametric Inaccessibility and Evidence Sufficiency is not clearly described—it’s unclear whether these evaluations are performed via prompting LLMs directly or through a separate automated verification strategy.
3. While the benchmark is well-motivated, the paper could further clarify its generalization scope, such as whether the hint-free design can extend beyond Wikipedia-style domains or integrate with open web environments to evaluate real-world reasoning robustness.

**Questions:**

Please refer to weaknesses.

---

> ### Author Response · Authors · 2025-11-24
> **Response to Reviewer yEz4**
>
> Thank you for your constructive feedback and positive assessment of our work. We're pleased that you find our hint-free benchmark design promising and our analysis insightful. We address your concerns below:
>
>
> ### **W1: Presentation Improvements**
>
> We acknowledge the presentation issues and will make the following improvements in the revision and revise the paper:
> - **Restructure key sections** to highlight main insights more prominently
> - **Redesign figures** for better clarity and visual emphasis
> - **Move critical findings** to more prominent positions rather than burying them in text
>
>
> ### **W2: Methodological Details on Parametric Inaccessibility and Evidence Sufficiency**
>
> We apologize that space constraints in the main text prevented a full elaboration of these verification protocols. We will add a dedicated section in the appendix detailing the following automated evaluation process:
>
> **1. Defining Knowledge Sufficiency**
>
> We define a question as "answerable" (or Knowledge Sufficient) if the union of **retrieved evidence ($E_{search}$)** and the model's **parametric knowledge ($K_{param}$)** fully covers the reasoning chain required to derive the answer.
>
> * **Condition:** The model effectively possesses the complete chain if, for every hop $h_i$ in the reference chain not covered by $E_{search}$, the model can supply the missing link via $K_{param}$.
>
> **2. The Automated Probing Process** To determine if the model possesses the specific missing knowledge ($K_{param}$), we employ an atomic probing strategy utilizing an **LLM-as-a-Judge** framework:
>
> * **Step A (Gap Identification):** We identify specific nodes or relations in the reasoning chain that were *not* successfully retrieved during the search phase.
>
> * **Step B (Atomic Probing):** For each missing link (e.g., node $A \to$ node $B$), we construct a targeted prompt to test the model's internal knowledge of that specific relation.
>   * *Example:* If the relation is "Kane Cornes has a brother named Chad Cornes," we query the model: *Please fill the blank: "Kane Cornes has brother ____ ?"*
>   * If the model correctly mentions *"Chad Cornes"* (without external context), we confirm it possesses this parametric knowledge.
>
> * **Step C (Consistency Verification):** An automated LLM judge compares the model's generated answer against the ground truth entity. If the judge confirms consistency, the hop is marked as "covered."
> If all necessary hops are covered either by the search buffer or this verified parametric knowledge, the instance is marked as **Knowledge Sufficiency**.
>
>
> ### **W3: Generalisation Beyond Wikipedia**
>
> We clarify that the "hint-free" design and our diagnostic framework are not tied to Wikipedia but are fundamentally portable methodologies.
>
> **1. Benchmark Core & Hint-Free Design**
>
> Our benchmark is defined by a "co-design" approach that evaluates reasoning by stripping away artificial artifacts (hints) and relying solely on the agent's ability to navigate a knowledge space. This design is agnostic to the underlying data source; it measures the universal capability of an agent to verify and chain information without explicit guidance.
>
> **2. Universal Applicability (The Co-Design Approach)**
>
> Our pipeline is highly extensible. It does not strictly require Wikipedia but rather generalizes to **any** domain that satisfies three fundamental conditions:
>
> 1. **A text corpus** with factual content
> 2. **An entity-level graph or link structure** over that corpus
> 3. **The ability to mask/obfuscate** specific entity mentions along paths
>
> We chose Wikipedia for this initial release because it provides:
> - Clean entity graphs for controlled evaluation
> - Sufficient complexity to expose reasoning failures
> - Reproducible experimental conditions
>
> **3. Beyond the Open Web**
>
> Consequently, this framework extends well beyond just "open web" scenarios. It can be instantiated on specialized proprietary datasets (e.g., enterprise knowledge bases) or dynamic news streams, allowing for the evaluation of real-world reasoning robustness in diverse, domain-specific environments.
>
> We will clarify in the revision that WebDetective's co-design principle is a general methodology applicable across domains, with Wikipedia serving as our initial controlled testbed rather than a fundamental limitation.

---

> ### Author Response · Authors · 2025-11-26
> **Request for your feedback**
>
> Dear Reviewer yEz4,
>
> Thank you for your thoughtful review. We have clarified your concerns: improving clarity and figures, expanding methodological details and benchmark’s generalisation scope.
>
> We hope these changes resolve the issues you raised. If so, we would be grateful if you could consider increasing the score. Please let us know if anything remains unclear.
>
> Thank you again for your time.
>
> The Authors

---

### Official Review · Reviewer_fQW4 · 2025-10-28

**Soundness:** 3
**Presentation:** 3
**Contribution:** 3
**Rating:** 6
**Confidence:** 2

**Summary:**

The paper introduces WebDetective, a benchmark for evaluating web agents on hint-free multi-hop question answering tasks. It also introduces a co-design evaluation environment to enable rigorous diagnostics. The evaluation contains multiple frontier systems and is comprehensive.

**Strengths:**

The writing is clear and easy to follow.

The problem formulation is novel, and the observation that existing benchmarks contain unrealistic hints and specifications is meaningful.

The diagnostic evaluation framework can help target different weaknesses of different systems, meaningfully characterize different systems under different categories, and provide insightful/actionable guidance to the development of future deep research systems.

**Weaknesses:**

1. The controlled Wikipedia sandbox, while enabling precise evaluation, creates artificial constraints that may not reflect real-world multi-hop reasoning scenarios. In practice, information is rarely hidden behind such strict sequential dependencies—multiple valid paths typically exist, and partial information can be retrieved from various sources. With an analysis of how performance correlates between sandbox and open-web settings, it could further enhance the validity and value of the benchmark.

2. The 200 questions are relatively small for establishing robust benchmark conclusions. It would be helpful to enhance the statistical rigor by including significance tests.
The heavy concentration on 2-3 hop questions (86%) limits insights about longer reasoning chains. Besides, the questions are synthetically constructed by blocking direct paths in existing QA pairs, which may not reflect naturally occurring multi-hop reasoning patterns.

3. The paper lacks analysis of domain diversity beyond question types, leaving unclear whether the benchmark targets diverse reasoning patterns or is dominated by specific domains (e.g., familial relationships).

**Questions:**

See Weakness.

---

> ### Author Response · Authors · 2025-11-24
> **Response to Reviewer fQW4**
>
> We thank the reviewer for their thoughtful and constructive feedback. We appreciate the reviewer's recognition of our novel problem formulation, clear presentation, and the diagnostic evaluation framework's value in characterising different systems and providing actionable guidance. We address each of the reviewer's concerns below:
>
> ### **W1: Controlled Wikipedia Sandbox vs Real-World Settings**
>
> We appreciate the reviewer’s concern regarding real-world complexity. However, we would like to clarify a misconception regarding the benchmark's design and highlight the extensibility of our framework.
>
> **1. Clarification on "Strict Sequential Dependencies"**
>
> Our benchmark **does not enforce** a single, strict reasoning path. While we utilize a constructed graph to guarantee solvability, our evaluation metric focuses on **Knowledge Sufficiency**.
>
> In practice, an agent is not penalized for finding alternative routes. Whether the agent retrieves partial information from diverse sources or utilizes parametric knowledge to bridge gaps, the benchmark counts the question as "answerable" as long as the union of retrieved evidence and internal knowledge is sufficient to derive the answer. Thus, multiple valid paths coexist and are correctly measured, mirroring real-world conditions where information is fragmented.
>
> **2. Extensibility to Open-Web Settings**
>
> Furthermore, our "co-design" approach is not strictly bound to Wikipedia. The methodology allows the sandbox to evolve into larger, open-web settings because the pipeline relies on only three fundamental assumptions:
>
> 1. A text corpus with factual content.
>
> 2. An entity-level graph or link structure over that corpus.
>
> 3. The ability to mask or obfuscate mentions of specific entities along chosen paths.
>
> Consequently, the exact same pipeline (path construction + masking + factorized evaluation) can be directly applied to more "open-web-like" corpora, such as curated news collections, scientific repositories, or domain-specific encyclopedias, ensuring the framework's value extends well beyond the current testbed.
>
>
> ### **W2: Dataset Size and Statistical Rigour**
>
> **Quality-Focused Construction Pipeline**
>
> The relatively small size (200 questions) is a **direct consequence of our strict construction and filtering pipeline**. For each candidate question-answer pair (q, a*) with evidence chain E = {e₁, ..., eₙ}, we enforce:
>
> 1. **Parametric Inaccessibility**: We query a strong verifier LLM with only question q. If it can reliably produce a* from parametric knowledge alone, we discard the question.
>
> 2. **Evidence Sufficiency**: We verify that LLM(q, E) = a*, ensuring the complete evidence chain enables correct answer generation.
>
> 3. **Evidence Necessity (Leave-one-out)**: For every i ∈ {1,...,n}, we probe with the incomplete set (q, E\{eᵢ}). If the verifier can still recover a* from any (n-1) evidence subset, the question is discarded. We retain only questions where the verifier succeeds with all n evidences but fails with every leave-one-out subset.
>
> The majority of automatically generated candidates are discarded during this rigorous process, followed by manual verification.
>
> **Statistical Considerations and Diagnostic Purpose**
>
> Our primary aim with WebDetective is to provide a **diagnostic benchmark** that exposes fundamental failure modes rather than ranking systems with marginal differences. The patterns we identify are substantial and consistent across all 25 evaluated systems—the gap between Knowledge Score (79%) and Generation Score (23%) for top models, the near-universal failure of refusal capabilities (< 30% Good Refusal F1), and the synthesis bottleneck (Knowledge Utilization peaks at 56%)—these are not marginal effects requiring large samples to detect.
>
> That said, we acknowledge the value of statistical reliability. Our binomial confidence analysis shows that with n = 200, we achieve a 95% confidence interval of ±6.9% (assuming 50% pass rate). While this is sufficient for identifying the large-scale failure patterns our benchmark targets, we plan to expand to ~385 samples in future releases to achieve ±5% intervals for finer-grained comparisons.
>
> **Challenge at Current Scale**
>
> Even with our 2-3 hop concentration (86% of questions), the benchmark remains **far from saturated**:
> - The highest Pass@1 is only 56% (O3-Pro)
> - State-of-the-art agents frequently fail to synthesize 2-3 pieces of evidence without hints
> - We observe substantial knowledge degradation even at these modest depths
>
> This indicates that 2-3 hint-free hops already constitute a significant challenge, validating our benchmark's diagnostic power at the current scale.

---

> ### Author Response · Authors · 2025-11-24
> **Response to Reviewer fQW4**
>
> ### **W3: Domain Diversity**
>
> WebDetective covers **diverse domains** to ensure broad evaluation coverage:
>
> | Domain | Percentage |
> |--------|------------|
> | Film, Television and Theatre | 20.0% |
> | History, Politics and Military | 18.0% |
> | Geography, Locations and Landmarks | 16.0% |
> | Music and Audio Production | 13.5% |
> | Sports and Competitions | 11.5% |
> | Biography, Genealogy and Education | 8.5% |
> | Literature, Comics and Pop Culture | 6.5% |
> | Science, Technology and Transport | 6.0% |
>
> This distribution demonstrates that WebDetective is **not skewed toward any single relation type** (such as familial ties), but instead spans a well-balanced set of domains essential for evaluating multi-hop reasoning. The controlled environment, intentionally modest dataset size, and broad domain coverage are deliberate design choices optimized for **diagnostic precision** rather than scale. Our goal is to provide an evaluation tool capable of clearly pinpointing where and why current systems fail at autonomous reasoning discovery—insights made possible only through this carefully controlled, purpose-built framework.

---

> ### Author Response · Authors · 2025-11-26
> **Request for your feedback**
>
> Dear Reviewer fQW4,
>
> Thank you again for your thoughtful and constructive feedback. We’ve incorporated all your suggestions in the revision, including: (1) analysis of sandbox–web performance correlations, (2) expanded dataset/domain diversity analysis, (3) added significance testing with stronger statistical reporting, and clarified multi-hop question construction.
>
> We hope these revisions address your concerns and strengthen the paper. If they do, we would be grateful if you might consider updating your score. If anything remains unclear, we’re happy to provide further clarification.
>
> Sincerely,
>
> The Authors

---

### Official Review · Reviewer_ZvCi · 2025-11-04

**Soundness:** 3
**Presentation:** 2
**Contribution:** 2
**Rating:** 6
**Confidence:** 3

**Summary:**

This paper presents WebDetective, a benchmark for evaluating web agents and RAG systems on hint-free multi-hop deep search tasks. It identifies two core issues in existing benchmarks: hint leakage in question design (Path- or Specification-Hinting) and single-metric evaluation that masks distinct failure modes. WebDetective co-designs hint-free questions with a controlled Wikipedia sandbox enforcing sequential reasoning through selective entity masking, and introduces a factorized diagnostic framework separating knowledge sufficiency, generation quality, and knowledge degradation. Evaluations on 25 models show that most retrieve sufficient evidence but fail in reasoning composition and calibrated refusal.

**Strengths:**

(1) The paper identifies a genuine gap in current deep-search benchmarks—most rely on linguistic hints that reduce reasoning to execution. The distinction between Path-Hinting, Specification-Hinting, and the proposed Hint-Free formulation is sharp and well grounded.

(2) The co-designed Wikipedia sandbox with selective entity masking is both creative and rigorous, ensuring that agents must discover reasoning paths rather than shortcut them. This design provides strong interpretability and enables precise failure attribution.

(3) Evaluating 25 state-of-the-art models provides broad coverage and meaningful insights. The taxonomy of behavioral profiles (e.g., Powerful but Overconfident, Synthesis Bottleneck) is particularly informative for understanding system limitations.

**Weaknesses:**

(1) The proposed EvidenceLoop baseline achieves modest improvements and is not clearly positioned relative to existing ReAct-style frameworks, making it less convincing as a substantive modeling advance.

(2) The controlled sandbox enforces a single reasoning path, which improves interpretability but departs from realistic web conditions where multiple reasoning routes may coexist. This trade-off limits ecological validity.

**Questions:**

(1) How do you ensure that human validators and verifying LLMs do not inadvertently rely on their own parametric knowledge when assessing whether questions are answerable only through the provided evidence chain?
(2) Could you clarify how EvidenceLoop differs conceptually and architecturally from prior ReAct or self-reflection frameworks? Is its improvement mainly from verification loops or memory structuring?

---

> ### Author Response · Authors · 2025-11-24
> **Response to Reviewer ZvCi**
>
> We thank the reviewer for their thoughtful and constructive feedback. We appreciate the reviewer's recognition of our novel problem formulation, clear presentation, and the diagnostic evaluation framework's value in characterising different systems and providing actionable guidance. We address each of the reviewer's concerns below:
>
> ### **W1: EvidenceLoop achieves modest improvements and is not clearly positioned relative to existing frameworks**
>
> We appreciate the reviewer's question about how EvidenceLoop differs from existing ReAct-style and Self-reflection agents. **EvidenceLoop is a generally applicable agentic workflow for deep search, guided by the failure modes revealed by WebDetective**, rather than a one-off diagnostic script tied to the benchmark. It introduces several key architectural differences:
>
> | Aspect | Standard ReAct | ReAct + Self-reflection | EvidenceLoop (ours) |
> |--------|---------------|------------------------|-------------------|
> | **Controller loop** | Single "Thought → Tool → Observation" loop until answer | Same loop, with occasional reflection/critique steps | Two phases: (i) exploration rounds to collect evidence; (ii) verification/decision over aggregated evidence |
> | **Memory** | Flat dialogue history (all thoughts + tool outputs) | Same history plus extra reflection messages | Structured evidence buffer: per-hop tuples (entity, snippet, page) plus controller state |
> | **Verification** | Often none | Reflections may critique but not tied to explicit evidence | Explicit verification loop using only evidence buffer; failure leads to refusal |
> | **Breadth & iterations** | Single search trajectory | Same as ReAct | Breadth = parallel trajectories per round; Iteration = explore→aggregate cycles |
> | **Refusal behavior** | Mostly implicit; models rarely refuse | Sometimes encouraged, but loosely specified | Refusal explicitly tied to verification: incomplete evidence → refusal not guessing |
>
>
> ### **W2: Single reasoning path limitation and ecological validity**
>
> We clarify that our benchmark **does not enforce a single reasoning path**, nor does it penalise agents for deviating from the provided chain.
>
> **1. Knowledge Sufficiency is assessed through three complementary mechanisms:**
>
> (1) Shortest Reference Chain Coverage. We first check whether all hops in the shortest reference chain were visited. For any missing evidence, we probe the model's parametric knowledge with targeted queries. If visited evidence plus verified parametric knowledge covers the complete chain, the instance achieves knowledge sufficiency.
>
> (2) Alternative Path Recognition. Agents may discover valid reasoning routes different from the reference chain. We collect all evidence the agent actually visited during search and feed this clean context to an LLM judge. If the judge can correctly answer the question from this visited evidence alone, the instance also achieves knowledge sufficiency—regardless of whether the reference chain was followed.
>
> (3) Efficient Parametric Knowledge Use. Our Search Score further credits agents that efficiently combine search with parametric knowledge. When an agent correctly answers using fewer (or equal) hops than the reference path while actively performing search, this demonstrates intelligent leveraging of internal knowledge to shortcut the reasoning process, and receives additional credit.
>
> This multi-faceted evaluation tracks what information the agent has, not which path it followed—combining controlled interpretability with flexibility to recognize diverse valid strategies.
>
> **2. An Example:**
>
> Consider a scenario where the **Reference Chain** (shortest path using corpus only) is $A \to B \to C \to \text{Ans}$. However, the agent, leveraging its own strategy or internal knowledge, takes an **Actual Path** of $A \to B \to D \to E \to \text{Ans}$.
>
> * **Traditional Metrics:** Might penalize the agent for missing node $C$.
>
> * **Our Metric (Knowledge Sufficiency):** We evaluate whether the state $\{A, B, D, E\}$ contains sufficient information to derive $\text{Ans}$.
>
>   * If the agent successfully retrieves evidence for $D$ and $E$, and these nodes provide the necessary context to answer the question (effectively bypassing the need for $C$), our metric marks the search process as successful.
>   * Similarly, if the agent utilises **parametric knowledge** to skip a retrieval step (e.g., moving directly from $A \to C$), the metric assesses the final information state sufficiency, ensuring the agent is credited for a valid solution. Thus, the sandbox maintains ecological validity by allowing diverse reasoning strategies while still providing a rigorous baseline for measurement.

---

> ### Author Response · Authors · 2025-11-24
> **Response to Reviewer ZvCi**
>
> ### **Q1: How to ensure validators don't rely on parametric knowledge**
>
> We clarify that there is a slight misunderstanding regarding our definition of "answerability." In our benchmark, a question is considered answerable based on **Knowledge Sufficiency**: as long as the **union** of *gathered evidence* and *parametric knowledge* completes the reasoning chain, the question is valid.
>
> Therefore, our validation protocol does not aim to *prevent* the use of parametric knowledge, but rather to explicitly **disentangle and measure** it. We ensure this through a two-step validation process for both human validators and verifying LLMs:
>
> 1. **Gap Analysis (Evidence-Based):** First, we assess the *searched evidence* to identify exactly which logical links (from Start $\to$ Ans) are missing or unsupported by the retrieval results alone.
>
> 2. **Parametric Probing:** We then perform a targeted prompt test to determine if the specific missing knowledge identified in Step 1 resides within the validator's (human or LLM) internal knowledge base.
>
> This design allows our benchmark to **explicitly allow and measure** both parametric-reliant and search-reliant reasoning paths, rather than artificially forcing a "blank slate" constraint that contradicts real-world LLM behaviour.
>
>
> ### **Q2: Conceptual differences from ReAct/self-reflection**
>
> Beyond the architectural differences discussed in the response to Weakness 1, the fundamental distinction is that EvidenceLoop is **diagnostically derived**. While ReAct and Self-Reflection focus on generic reasoning steps, **WebDetective's diagnostics explicitly revealed specific failure modes** in those frameworks—namely synthesis failures, poor calibration, and knowledge degradation.
>
> Consequently, EvidenceLoop was engineered to target these specific weaknesses. The combination of parallel exploration, structured evidence tracking, iterative refinement, and verification-tied refusal is not merely a heuristic assembly but a direct response to the challenges identified by our diagnostic framework.
>
> To quantify the influence of the verification and memory components, we conduct ablations. The results show that both mechanisms primarily address *knowledge-related shortages*, with memory having the strongest effect on knowledge discovery.
>
> | Model | Search Score | Knowledge Util F1 |
> |-------|--------------|-------------------|
> | *EvidenceLoop (DeepSeek-R1-0528)* | *71.5* | *25.0* |
> | – w/o Memory | 63.0 | 24.38 |
> | – w/o Verification | 68.6 | 23.46 |
>
> We find that removing memory leads to the largest reduction in search performance (−8.5), while removing verification decreases both search (−2.9) and utilization (−1.54).
> Overall, *memory-driven working-context structuring is the main contributor to mitigating knowledge-related shortages*, with verification providing additional support for both retrieval quality and synthesis stability.

---

> ### Author Response · Authors · 2025-12-01
> **Supplementary W1 Experiments**
>
> To further demonstrate that *EvidenceLoop meaningfully improves over ReAct-style methods—particularly on context degradation and limited search coverage*—we additionally evaluate it on two different model architectures (DeepSeek-R1-0528 and GLM-4.5-Air). This tests whether the gains generalise beyond a single backbone.
>
> Across both architectures, EvidenceLoop yields *consistent improvements* in knowledge discovery and utilisation:
>
> - *Search coverage:* +3 to +6 Search Score
> - *Knowledge utilization:* +10% to +53% relative F1
> - *Overall task performance:* +4–8 Pass@1
>
> | Model | Knowledge Suff | Search Score | Knowledge Util F1 | Pass@1 |
> |-------|----------------|--------------|-------------------|--------|
> | DeepSeek-R1-0528 | 61.0 | 65.5 | 16.36 | 20.5 |
> | *EvidenceLoop (DeepSeek-R1-0528)* | *68.0 (+7.0)* | *71.5 (+6.0)* | *25.0 (+53% rel.)* | *28.5 (+39% rel.)* |
> | GLM-4.5-Air | 55.5 | 60.5 | 17.97 | 19.0 |
> | *EvidenceLoop (GLM-4.5-Air)* | *62.0 (+6.5)* | *63.5 (+3.0)* | *19.7 (+10% rel.)* | *23.5 (+24% rel.)* |
>
> These cross-architecture results confirm that EvidenceLoop’s iterative retrieval loop provides *generalizable gains* beyond standard ReAct—strengthening both search coverage and evidence integration.

---

### Author Response · Authors · 2025-12-01
**General Response for Area Chair: WebDetective**

**Dear Area Chair,**

We sincerely thank all reviewers for their constructive feedback throughout the discussion. Their valuable suggestions and proposed experiments have helped us improve our work and clarify our contribution. Below, we provide a concise summary of the key points.

---

## **Overview**

We introduce **Hint-Free Multi-Hop QA** as a new evaluation paradigm. Existing benchmarks embed hints—Path-Hinting (PH) or Specification-Hinting (SH)—that reduce the task to executing pre-specified paths rather than discovering them. To our knowledge, WebDetective is the first to:

1. **Enable hint-free deep search evaluation** (**S1**) for assessment of systems’ ability to autonomously conduct information retrieval and reasoning without any form of hint (PH or SH)
2. **Provide factorized diagnostic metrics** (**S3**) separating knowledge sufficiency, generation quality, and refusal calibration—enabling fine-grained failure attribution and revealing a taxonomy of behavioural profiles (**S4**)

Our evaluation of 25 frontier models (**S2**) shows models achieve much higher knowledge sufficiency than generation quality, indicating synthesis and calibrated refusal, instead of retrieval, is the primary bottleneck. These findings demonstrate that current systems perform poorly when required to discover reasoning paths autonomously without hints.

---

## **Commonly Recognised Strengths**

We summarised the following strengths highlighted by reviews:

| Code | Strength | Recognised by |
| :---- | ----- | ----- |
| S1 | Identification of hinting issues (Path-Hinting, Specification-Hinting) and the hint-free problem setting are novel and well-motivated | ```ZvCi```, ```fQW4```, ```yEz4```, ```rUPx``` |
| S2 | Comprehensive evaluation across 25 frontier models with insightful analysis | ```ZvCi```, ```fQW4```, ```yEz4```, ```rUPx``` |
| S3 | Factorised evaluation framework provides actionable and fine-grained analysis for targeting system weaknesses | ```ZvCi```, ```fQW4```, ```yEz4``` |
| S4 | Taxonomy of behavioural profiles (e.g., Powerful but Overconfident, Synthesis Bottleneck) is informative for understanding system limitations | ```ZvCi```, ```fQW4```, ```yEz4``` |
| S5 | Co-designed sandbox with selective masking is creative and rigorous | ```ZvCi```, ```rUPx``` |

---

## **Summary of Reviewer Feedback & Corresponding Actions**
We have summarised all points raised by reviewers and addressed all concerns (C1–C6) through **both** clarifications in the discussion and revisions to the paper. For C3, we additionally conducted new experiments on 2 base models with additional component-wise ablation studies.
| Code | Concern | Raised by | Our Action |
| ----- | ----- | ----- | ----- |
| C1 | Whether the benchmark supports multiple reasoning paths | ```ZvCi```, ```fQW4```, ```rUPx``` | Clarified that benchmark accommodates multiple valid paths—we track what information the agent has, not which path it followed |
| C2 | Whether the framework extends beyond Wikipedia | ```fQW4```, ```rUPx```, ```yEz4``` | Clarified co-design methodology generalises to any corpus with entity-level structure (news, scientific repositories, enterprise KBs) |
| C3 | EvidenceLoop improvements and architectural differences from ReAct | ```ZvCi```, ```rUPx``` | Detailed architectural differences; conducted experiments on 2 additional base models with component-wise ablations |
| C4 | Asks how we evaluate knowledge-sufficiency given the model's parametric knowledge? | ```ZvCi```, ```yEz4``` | Clarified we allow models to use their parametric knowledge and we measure it by probing the models with formatted prompts for each evidence |
| C5 | Dataset scale and domain diversity | ```fQW4``` | Explained rigorous filtering pipeline; patterns reported are large-scale effects; added analysis showing 8 diverse domains |
| C6 | Presentation and methodological details | ```yEz4``` | Revised figures; added probing protocol details in appendix |
| C7 | Which base LLM for EvidenceLoop | ```rUPx``` | Clarified DeepSeek-R1; now explicit in text |

---

We appreciate your pivotal role in this final assessment. Thank you for your time and effort for the community.

Sincerely,
Authors of Submission 13896

---

### Meta-Review · Area_Chair_6Hch · 2026-01-07

**Summary:**

This paper introduces WebDetective, a multi-hop question answering dataset that is designed to minimize the amount of hints given to an agent about how to search for an answer. Previous multi-hop QA datasets like HotpotQA and BrowseComp contain fairly explicit search paths within the query itself (“Who is the husband of the stepmother of the brother of Kane Cornes?”). In contrast, WebDetective would just ask “Who is the father of Kane Cornes?” and allow the model itself to figure out that the only path to the answer is the complex multi-hop path in the HotpotQA query. Given the strong complexity filters used in its construction, the dataset consists of only 200 valid questions. Nevertheless, it is able to demonstrate through these examples that most frontier models fail to solve these simple (2-3 hop questions) successfully. The authors also propose a more complex prompting method called EvidenceLoop that improves the performance of open-source models modestly.

All concerns presented by the reviewers were adequately addressed and the original reviews were strong (6,6,6,6). I believe that this paper should be accepted.

My only disagreement with the authors is that extending this framework to other scenarios and domains is non-trivial and I recommend that this is stated alongside the three criteria required for generalization.

**Reviewer Concerns:**

R1: 6,3
- EvidenceLoop achieves modest improvements and is not contextualized in terms of ReAct
	- Well-structured comparison to ReAct and ReAct + Self-Reflection. EvidenceLoop is a complex system with different components for controlling models, storing memory, verifying information, controlling breadth and iteration and refusing to answer.
	- An ablation was also created to understand the effects of memory and verification components within EvidenceLoop
- Multiple paths
	- Valid paths that are different from the shortest reference path are still recognized as correct based on an LLM-as-a-judge.
- Parametric memory for verification?
	- The “Knowledge Sufficiency” metric is explicitly designed such that if an agent is relying on their parametric knowledge, the validator needs to probe it to understand if that knowledge is “present” in its parameters. This is an interesting but approximate proxy for determining whether a model “knows” and can leverage a specific fact.

R2: 6,2
- Wikipedia sandbox is too constrained
	- Knowledge sufficiency allows for multi-path settings
	- The authors claim that this methodology can be used in open web settings that have entity-level graphs or link structures. This is not a very convincing argument since this process would be much more challenging and noisy in settings that do not have explicit entity links like Wikipedia.
- Dataset is small and the questions are short and not realistic
	- The dataset is small because of the strict filtering enforced to keep questions challenging.
	- The multi-hop questions are short but still very challenging (56% success of o3-pro)
- Significance testing is necessary
	- Most of the performance trends for different model types have very large differences which do not really require significance testing to confirm but the authors still provide some significance testing for completeness.
- Analysis on reasoning pattern and domain distribution is missing
	- The authors added the distribution of domains. I would like to see much more examples than the one provided in the paper, it makes it seem like the only question of this sort is the family question.

R3: 6,4
- Presentation could be improved
	- The authors promise to improve the paper’s presentation
- Some parts of methodology are under explained
	- The Knowledge Sufficiency process is explained in detail in the response.
- Can this process be generalized outside of Wikipedia?
	- The same response given for R2’s Wikipedia question. I still believe this generalizability argument is not very convincing since “entity-level graph or link structure” are hard to find in other scenarios. This would require a new design for other scenarios and is not trivial.

R4: 6,4
- Wikipedia sandbox is to constrained
	- Same answer as R2 and R3.
- Modest improvements
	- The authors argue that the improvements are significant and further demonstrate this by adding a GLM-4.5-Air experiment that shows similar trends. More models would likely be needed to convincingly demonstrate that EvidenceLoop provides significant improvements (especially given the size of the dataset), however, the paper’s main contribution is showign failure modes of frontier models in simple deep research settings so this issue is secondary.

**Reviewer Scores:**

- ZvCi 6 -> 6
- fQW4 6 -> 6
- yEz4 6 -> 6
- rUPx 6 -> 6

---

### Decision · Program_Chairs · 2026-01-26

Accept (Poster)